# Analysis of mobile monitoring data from the microAeth® MA200 for measuring changes in black carbon on the roadside in Augsburg

Xiansheng Liu[1,2], Hadiatullah Hadiatullah[3], Xun Zhang[4,5], L. Drew Hill[6], Andrew H. A. White[6,7], Jürgen Schnelle-Kreis[1], Jan Bendl[1,8], Gert Jakobi[1], Brigitte Schloter-Hai[1], Ralf Zimmermann[1,2]

[1]Joint Mass Spectrometry Center, Cooperation Group Comprehensive Molecular Analytics, Helmholtz Zentrum München, German Research Center for Environmental Health, Ingolstädter Landstr. 1, 85764 Neuherberg, Germany

[2]Joint Mass Spectrometry Center, Chair of Analytical Chemistry, University of Rostock, 18059 Rostock, Germany

[3]School of Pharmaceutical Science and Technology, Tianjin University, 300072 Tianjin, China

[4]Beijing Key Laboratory of Big Data Technology for Food Safety, School of Computer Science and Engineering, Beijing Technology and Business University, 100048 Beijing, China

[5]Key Laboratory of Resources Utilization and Environmental Remediation, Institute of Geographical Sciences and Natural Resources Research, Chinese Academy of Sciences, 100101 Beijing, China

[6]AethLabs, San Francisco, CA, USA

[7]Yale School of Medicine, New Haven, CT, USA

[8]Institute for Environment Studies, Faculty of Science, Charles University, Prague, Czech Republic

*Correspondence to*: Xun Zhang (zhangxun@btbu.edu.cn); Jürgen Schnelle-Kreis (juergen.schnelle@helmholtz-muenchen.de).

**Abstract.** The portable microAeth® MA200 (MA200) is widely applied for measuring black carbon in human exposure profiling and mobile air quality monitoring. Due to its relatively new in the market, the field lacks a refined assessment of the instruments performance under various settings and data postprocesing approaches. This study assessed the mobile real-time performance of the MA200 in an urban area, Augsburg, Germany. Noise reduction and negative value mitigation were explored via different data postprocessing methods (i.e., local polynomial regression (LPR), optimized noise reduction averaging (ONA), and centered moving average (CMA)) under common interval time (i.e., 5, 10, and 30 s). After noise reduction, the treated-data were evaluated and compared by (1) the amount of useful information attributed to microenvironmental characteristics retained; (2) relative number of negative values left; (3) reduction and retention of peak-samples; and (4) the amount of useful signal retained after correction for local background conditions. Our results identify CMA as a useful tool for isolating the central trends of raw black carbon concentration data in real time while reducing non-sensical negative values and the occurrence and magnitudes of peak-samples that affect visual assessment of the data without substantially affecting bias. Correction for local background concentrations improved the CMA treatment by bringing nuanced microenvironmental changes into more visible. This analysis employs a number of different postprocessing methods for black carbon data, providing comparative insights for researchers looking for black carbon data smoothing

approaches, specifically in a mobile monitoring framework and data collected using the microAeth®
series of aethalometers.

Keywords: Black carbon; Mobile measurement; Noise reduction; Peak-sample; Background correction

**1 Introduction**

Black carbon particulate matter with size ranging from 0.01 to 1 μm (Zhou et al., 2020), is a pollutant
comprising a range of carbonaceous materials produced by the incomplete combustion of fossil fuel
and biomass containing carbon (Goldberg, 1985), and is suspected of exerting significant impact on
health (Anenberg Susan C. et al., 2012; N. A. H. Janssen et al., 2011; Nichols et al., 2013).
Subsequently, it also has an important role in climate systems due to its strong radiative forcing
potential (Kutzner et al., 2018, Sadiq et al., 2015). The International Agency for Research on Cancer
(IARC) has classified black carbon as a 2B carcinogen, while researchers have linked black carbon
exposures to cardiovascular, respiratory, and neurological diseases (e.g., Nichols et al., 2013). However,
the hyper-local nature of air quality among small-scale urban blocks is difficult to characterize with the
existing monitoring networks which typically rely on fixed monitors (Apte et al., 2017), especially for
on-road concentrations.

Although some progress has been made in the study of black carbon monitoring, however, many
studies are limited for mobile monitoring data. In the previous studies, Hegler et al. (2011) and Van
den Bossche et al. (2015) evaluated the optimized noise reduction averaging (ONA) for postprocessing
mobile monitoring data. However, due to the high spatial heterogeneity of black carbon, the ONA
algorithm may ignore important microenvironmental effects and lead researchers to perhaps incorrectly
conclude that resolution of microenvironmental source information cannot be determined from their
data. One manufacturer of aethalometers suited for mobile monitoring (by their size, weight, and
battery characteristics), AethLabs (San Francisco, CA, USA) offers additional forms of data
postprocessing (i.e., noise reduction), that given their accessibility and potential direct application to
the field and may be representative of methods, including the local polynomial regression (LPR) and
centered moving average (CMA) algorithms. The interpretation accuracy of data analyzed and reported
upon in black carbon mobile monitoring study can be increased by assessing the relative performance
of these methods to each other and to ONA.

An AethLabs instrument, the microAeth® MA200 (MA200; AethLabs, San Francisco, CA, USA) was
recently developed for measuring personal exposures to black carbon, ambient and vertical profiles of
black carbon concentrations, and indoor emissions concentrations of black carbon, among other black
carbon phenomena. The MA200 continuously collects aerosol particles on a filter and measures the
optical attenuation (ATN) at 5 wavelengths (880, 625, 528, 470, and 375 nm) with a data collection
time-base as frequent as 1 Hz. The cross-spectrum measurement provides insight into the composition
of light-absorbing carbonaceous particles and allows to distinguish among the different optical
signatures of various combustion sources such as fossil fuel (primarily diesel), biomass, and potentially

tobacco combustion (Helin et al., 2018). This instrument supports the DualSpot® loading compensation method, which corrects the optical loading effect (Virkkula et al., 2007) and provides more additional information about aerosol optical properties. In our study, the equivalent black carbon (eBC), the preferred term for describing black carbon assessed with mass absorption cross-section (MAC) facilitated optical absorption methods (Petzold et al., 2013), was used when targeting
quantitative values.

The raw data outputted by the MA200 at high frequencies (e.g., 1 Hz) can exhibit noise that obscure nuanced signals surrounding the central tendency of the data, increasing the difficulty of analysis in mobile settings or during rapidly changing micro-environmental characteristics. These negative values usually contain valid information required for noise reduction or smoothing, and so simply removing
them may result in bias. Noise reduction of the raw data without direct removal of negative values is thereby recommended to enhance data quality and temporal resolution (Liu et al., 2020). Moreover, high-time resolution measurements of air quality at roadside are susceptible to one-off events (e.g., the occasional passing of heavy-duty diesel vehicles or the stochastic passing of a cigarette smoker) that may not represent the general context of the street in study. This may lead to overestimation of eBC
levels when averaged over time/space as they introduce peaks in the dataset. In addition, when the sampling equipment traverses from highly-polluted to a low-polluted area, such as a park, the instrument produces strong negative values due to the measurement principle of the instrument and the strength of the pollution gradient between microenvironments. Therefore, the noise reduction method should also be evaluated based on the retention of actual peak-samples concentrations and number of
peak-samples associated with identifiable sources of pollution.

In addition, air pollution concentrations at a specific time and place may consist of two primary aspects: contributions from local source emissions and a background concentration (Tan et al., 2014). Background concentrations, especially the high background concentration of typical pollution events (such as haze), can obscure the contribution of local sources of pollution (Van et al., 2013; Van den
Bossche et al., 2015). Moreover, real-time changes in local sources, meteorology, and regional transport cause changes in the pollution background (Brantley et al., 2014), which will affect the comparability of measurements over different time periods, even at different times on the same days (Li et al., 2019). Therefore, we employed background concentration values to evaluate the noise reduction in mobile data after postprocessing and to provide a better assessment for local sources
contribution of air pollution to measured concentrations.

In this study, the application of several common methods for postprocessing black carbon data to improve reliable mobile measurements at high frequencies, including ONA (Hagler et al., 2011), LPR, and CMA, was assessed in the urban city. The postprocessing assessments data were focused on the microAeth® MA200. The quality of each noise reduction approach was assessed by analyzing
post-processed data under the following criteria: (1) retention of detailed information attributed to microenvironmental characters; (2) relative number of negative values remained; (3) reduction and

retention of peak-samples; and (4) retention of detailed information on microenvironmental characters after background correction.

## 2 Methods

### 2.1 Instrumentation

In this study, seven MA200 portable black carbon monitors (MA200-0051, MA200-0053, MA200-0059, MA200-0060, MA200-0155, MA200-0153, MA200-159) (microAeth® MA200; AethLabs, San Francisco, CA, USA) were used simultaneously to measure black carbon levels at the city center under different interval times (5 s, 10 s, and 30 s). The MA200 measures optical ATN from black carbon on a filter across 5 optical wavelengths: infrared, red, green, blue, and ultra-violet (880, 625, 528, 470, and 375 nm, respectively). Measurement of optical ATN at 880 nm characterizes the eBC concentration. The detection limit of the MA200 is reported at 30 ng eBC/m$^3$ and it notifies concentrations at the resolution of 1 ng/m$^3$ (AethLabs, 2018). In mobile monitoring, the MA200 can be used to estimate personal exposure and quantify eBC mass concentrations in different microenvironments. It can be used to identify the hot spots and to quantify black carbon levels on roads and highways as well as in various other mobile environments (Apte et al., 2011, Dons et al., 2012, Madueño et al., 2019) including bicycles (Wójcik et al., 2014, Samad and Vogt, 2020), trains (Andersen et al., 2019), and airplanes (Kim et al., 2019). The device can also be applied in long-term stationary monitoring, vertical profiling, and atmospheric measurement with unmanned aerial vehicles (Cao et al., 2020, Chiliński et al., 2018, Pikridas et al., 2019), balloons (Ferrero et al., 2016, 2014, 2011, Markowicz et al., 2017, Samad and Vogt, 2020), community monitoring, indoor air quality monitoring, and the assessment of personal exposure and related health effects (Isley et al., 2017). In order to reduce the noise concentration of the data obtained with high time resolution, smoothing algorithms can be used.

AethLabs offers tools for applying several noise reduction algorithms to MA-series device data on its website (https://aethlabs.com [note: a free account is required]). To evaluate the relative performance of MA200, this study analyzed black carbon data collected from multiple MA200 devices, identified individually by serial numbers. Comparative measurements of the MA200 and a stationary Aethalometer (AE33, Magee Scientific, Berkeley, USA) taken approximately for 30 to 60 min between walks showed a good agreement (Pearson's r =0.933) (Liu et al., 2021). In addition, it is worth noting that when the AE33 was used for monitoring black carbon at the same time as the MA200, the AE33 was placed in container, while MA200 was used outdoor (in the stroller) during the individual walks, which may have different relative humidity and temperature. This phenomenon did not influence the consistency of eBC concentration measured with both instruments. Information about the date, duration, and time resolution (time base) of each MA200 device are summarized in Table 1. To demonstrate the unit-to-unit comparability between the MA200 units, we performed intercomparisons at fixed monitoring stations (Table S1) and during collocated mobile measurements (Fig. S2). No wavelength

dependence was observed between different instruments for fixed and mobile monitoring measurements.

**Table 1** Measurements of black carbon by different MA200 devices.

| Measurement number | Date (dd/mm/yyyy) | Serial number | Start time (hh:mm:ss) | End time (hh:mm:ss) | Time base (s) | Site |
|---|---|---|---|---|---|---|
| 1 | 27/09/2018 | MA200-0051 | 10:29:10 | 13:38:20 | 10 | |
| 2 | 15/11/2018 | MA200-0059 | 11:53:42 | 16:13:12 | 10 | |
| 3 | 16/11/2018 | MA200-0053 | 11:34:06 | 16:33:56 | 10 | Augsburg, Germany |
| 4 | 26/08/2019 | MA200-0060 | 11:01:56 | 15:44:46 | 10 | |
| 5 | 21/02/2020 | MA200-0155 | 10:00:10 | 13:10:00 | 5 | |
| 6 | 21/02/2020 | MA200-0153 | 10:00:10 | 13:10:00 | 10 | |
| 7 | 21/02/2020 | MA200-0159 | 10:00:10 | 13:10:00 | 30 | |
| 8 | 24/11/2020 | MA200-0059 | 09:40:57 | 11:09:07 | 10 | |
| 9 | 01/12/2020 | MA200-0051 | 13:29:05 | 15:19:00 | 5 | Munich, Germany |
| 10 | 18/12/2020 | MA200-0051 | 14:39:30 | 15:19:30 | 30 | |

## 2.2 Study design and routes

The MA200 instrument is able to measure black carbon in 1 s, 5 s, 10 s, 30 s, 60 s, and 300 s interval times. The 1 s time base exhibits the most challenging interpretation because of poor signal to noise ratio especially at low concentrations, which is similar to other optical black carbon monitors (Hagler et al., 2011). Therefore, 1 s measurement resolution may be most useful when sampling in high concentration environments, performing direct emissions testing and requiring high time resolution for application. However, the eBC average concentration is low in the city center of Augsburg, Germany, (measured at 2.62 μg/m$^3$ in winter by Gu, (2012)) thus we did not use the 1 s time base. Moreover, 60 s and 300 s are too long distance for mobile monitoring, which may affect the accuracy of the spatial variation of pollutants, hence both time bases were also not selected in this study. In order to better understand at which interval time of sampling might be most useful in this context, mobile measurements at low eBC concentrations, three MA200 devices were used in parallel to measure eBC concentrations with the interval times of 5 s, 10 s, and 30 s (Measurement numbers 5-7 in Table 1).

To account for the different land use types of the microenvironments, a fixed walking route within the center of the city was determined. Wherever possible, the mobile measurements were carried out on the right side of the road simulating people's common habits (driving and walking on the right side in Germany). All walks along the route were conducted on weekdays, with clear skies and calm winds to avoid misrepresentation of typical urban exposure conditions. The route started from Augsburg University of Applied Sciences (UAS) and continued approximately 14 km for 3 h average walking time, passing through different types of land use to ensure that different microenvironments were represented the entire areas and the validity of the results (Fig. S1). Meanwhile, as performed in our previous study (Liu et al., 2021), we divided the monitoring route into four microenvironment groups

in Augsburg, including high traffic flow (H_Traffic, average 500-1000 vehicles/h), medium traffic flow (M_Traffic, average 200-500 vehicles/h), low traffic flow (L_Traffic, average 1-200 vehicles/h), and park area (N_Traffic, average 0 vehicles/h), according to the actual traffic density examined during the daytime and determining from the traffic flow observed by street views.

Briefly, the study was consisted of the following phases, (1) collecting raw black carbon data using the sampling instruments (MA200); (2) smoothing the acquired raw black carbon data under different postprocessing methods (i.e., noise reduction); (3) comparing the noise reduction data based on the detail change of value characters and number of negative value; (4) following the peak-samples identification by coefficient of variation (COV) approach and (5) following the background estimation and correction by thin plate regression spline (TPRS) approach; (6) finally, selecting the best noise reduction approach.

### 2.3 Instrumentation preparation

The instruments were prepared and adjusted in our laboratory before each walk, consisting of "zero" calibration checks, the examination of the MA200 filter cassette, battery, GPS, and memory checks. Flow calibrations were adjusted with a factory-calibrated flow meter (Alicat Scientific, Inc. Tucson, AZ, USA).

### 2.4 Postprocessing methods

The relative utility of the different postprocessing methods is determined by (1) the ability to perceive nuanced differences between microenvironmental pollution characteristics after after noise reduction; (2) the relative number of negative eBC values remained; (3) the reduction and retention of peak-samples; and (4) the ability to perceive nuanced differences between microenvironmental pollution characteristics with the noise-reduced data after background correction. These methods include ONA, LPR, and CMA.

#### 2.4.1 ONA (optimized noise reduction averaging)

The principle of the ONA is based on the time series of three parameters in the original observation data, namely the observation time, the original eBC concentration, and the optical ATN, as specifically described by Hagler et al. (2011). Briefly, a $\Delta$ATN threshold is manually set to prevent the algorithm from recalculating eBC until a certain amount of ATN has been detected (e.g., enough black carbon has deposited on the filter to "confidently" calculate an eBC concentration). The aims to reduce erroneous and spurious estimation by dynamically extending the effective sample time-base, hence, there is sufficient ATN to significantly reduce the error effects of instrument noise. This effective time-base will be longer in low concentrations than at higher concentrations and, hence, *no* negatives and less eBC noise will be reported. When using ONA algorithm, this $\Delta$ATN threshold needs to be manually assigned. Hagler et al., (2011) implemented a $\Delta$ATN threshold of 0.05 to postprocess data from a fixed monitoring site. However, when applied to MA200 data, a $\Delta$ATN threshold of 0.05 results in a very smooth curve and may obscure more information than is necessary to provide a usefully smoothed

curve. For this reason, a lower ΔATN threshold of 0.01 was selected for the mobile measurement data of our study (Figure S3).

### 2.4.2 LPR (local polynomial regression)

The LPR algorithm is a non-parametric tool similar to a moving average, but it operates on polynomial regression rather than simple averaging (Masry, 1996, Breidt and Opsomer, 2000, Kai et al., 2010). In LPR, the number of points across which to smooth must be manually identified. This value should be chosen to balance effective smoothing of the measured values and the sensitivity required to provide spatial resolution in mobile measurements (e.g., distance over which the average was taken). The distance resolution was chosen at approximately about 100 m. Assuming the sampling speed is 1.3 m/s, when the interval time is 5 s, 10 s, and 30 s, the smoothing number of points are 15, 7, and 3, respectively.

### 2.4.3 CMA (centered moving average)

The CMA algorithm is a smoothing technique used to make the long-term trends of a time series clearer (Easton and McColl, 1997). Unlike a simple moving average, CMA has no shift or group delay in the data processing, as it incorporates data from both before and after the datapoint that is being smoothed. The smoothing number of points was determined as previously described in the LPR algorithm, assuming a sampling speed of 1.3 m/s.

### 2.5 Comparison analysis after noise reduction approach

#### 2.5.1 The nuance of microenvironmental characters and the proportion of negative values.

After postprocessing data, the character change of the treated data is used as criterion to select the best method. In this regard, when the treated data provide more detailed microenvironmental characters, the data reflect the actual situation of air pollutants and facilitate the identification of pollution sources. However, if the microenvironmental characters is less detailed, it may hinder to identify the pollution source. Therefore, more detailed microenvironmental features contributed more accurate information. In addition, the number of remaining negative values is determined as another criterion to propose the best method. And, the method with the smallest proportion of the negative values is selected as the best method. The proportion of negative values remaining was calculated as the number of negative values divided by the total sample size.

#### 2.5.2 Peak-sample identification

An earlier study by Brantley et al. (2014) compared several methods for identifying and eliminating peak-samples in mobile air pollution measurements. These include identifying samples outside of a threshold based on a median produced using road segmentation, an α-trimmed arithmetic average (Van den Bossche et al., 2015), a running coefficient of variation (COV) (Hagler et al., 2012), an estimate of background standard deviation (Drewnick et al., 2012), a running low 25 % quantile (Choi et al., 2012)

and 3 times the standard deviation (Wang et al., 2015). The formula for the running method used in this analysis is previously described by Hagler et al. (2012) with minor modification (Eq. 1):

$$COV_t = \frac{\sqrt{\frac{1}{7}\sum_{i=t-3}^{i=t+3}(x_i - \bar{x})^2}}{\bar{x}_{all}}$$

(1)

where $COV_t$ is the 70 s sliding COV of the t-th eBC sample under a 10s timebase (representing 30 s prior to the sample, the sample, and 30 seconds after the sample), $x_i$ is the i-th eBC sample, $\bar{x}$ is the average of the t-th eBC sample and the three samples before and after it, and $\bar{x}_{all}$ is the average of all eBC data in one experiment. The 99th quantile of the 70 s sliding COV of all eBC data is used as the threshold for determining "peak-sample". The eBC samples that are greater than this threshold are flagged as peak-samples along with the eBC samples 3 data points before and after. However, under different time bases (e.g., 5 s, and 30 s), the sliding COV of the t-th black carbon sample are different. Accordingly, the COV equation is required for modification under different time base.

To calculate the reduction value of peak-samples, the number of peak-samples was calculated before and after postprocessing data, and the difference value was obtained. Then the change in the number of peak-samples was divided by the total number of peak-samples before postprocessing data to calculate the proportion of peak-samples values. After noise reduction, we compared the reduction values and the number of peak-samples to further evaluate postprocessing methods. In short, if the reduction value of peak-samples is high, the treated data has a high peak noise reduction without removing the numbers of peak-samples. Therefore, the method with high reduction value of peak-samples and retaining the number of peak-samples after postprocessing is considered as the better method.

### 2.5.3 Background estimation and correction

The ability of a processing method to adequately remove the estimated background concentration was used to evaluate which method provides the most useful information related to microenvironmental effects. A noise reduction method that appears to better facilitate background estimation and correction (as described below calculated from noise-reduced data via a defined background estimation and evaluation approach) is assessed to select a better postprocessing method.

Background correction methods include the single sample standardization method, the sliding minimum method, the linear regression postprocessing method, and the spline (of minimum) regression postprocessing method. Brantley et al. (2014) suggests that a thin plate regression spline (TPRS) method can reliably evaluate the background value of mobile measurements, and used to examine the "useful" information in the noise-reduced data (i.e. non-spurious, non-background pollution trends). Briefly, the TPRS approach includes three steps: first, the noise reduction data of pollutant was processed by a 30 s moving average; second, the results of the 30 s moving average were sequentially processed by the specified time window (i.e., 5 and 10 min), and the position of the minimum sample of pollutant concentration was identified in each window; and finally, thin-plate spline regression was

used to fit the sample of minimum pollutant concentration obtained in the previous step, then the background concentration at each time point was obtained.

**3 Results and discussion**

The average eBC concentrations of raw, ONA-processed, LPR-processed, and CMA-processed data (Measurements 1-10) monitored by all instruments were compared in this study (Table S2). The results show that the three postprocessing methods accounted of ±1 % bias from the average of raw concentrations. This indicates that the average concentration under each postprocessing method did not affect the average concentration of the raw unprocessed data.

**3.1 Postprocessing data under different interval time**

As shown in Figure 1, three MA200s were used at the time bases of 5 s, 10 s, and 30 s. The proportion of negative values in the raw data collected under different time base of was 42.1 %, 37.6 %, and 30.5 %, for 5 s, 10 s, and 30 s, respectively (Fig. 1a, Table 2, Fig S4a). Following that, the raw data were processed using ONA, LPR, and CMA (Fig. 1b, 1c, and 1d).

In the 5 s time base, the eBC values changed very rapidly (Fig. 1a), and the ONA processing of the data resulted in only one value (which was negative) (Fig. 1b). Thus, the microenvironmental characters of the eBC concentration was not reproduced. We found all $\Delta$ATN (ATN$_{t(0)+\Delta t'}$−ATN$_0$) data were negative in the raw data collected at 5 s, which, according to the ONA method described above, resulted in only a single value. In short, after the first measurement, the $\Delta$ATN threshold (which is positive) for calculating the next value was never reached. The first value was likely a negative value due to a combination of instrument noise, coincidence, and a low background concentration (i.e., low baseline instrument signal), which is consistent with both the raw data measurements and the typical low eBC concentrations in the city center of Augsburg, Germany (Gu, 2012). It is unclear why $\Delta$ATN remained negative, but, given the long series of low concentration vales in the beginning of the sample and the initial negative measurement, it is possible that the summed $\Delta$ATN became increasingly negative as a result of the initial negative $\Delta$ATN measurement. The subsequent measurements at low-concentration did not exceed the magnitude of the initial negative $\Delta$ATN value. Under these conditions, a cumulative negative sum of $\Delta$ATN would prevent the positive $\Delta$ATN threshold from being achieved at all. If true, this phenomenon highlights one potential weakness of the ONA algorithm, such as difficulty registering a signal under low concentrations and requires further investigation of the conditions under which ONA is truly unbiased. At any rate, the observed phenomenon prevented the use of ONA in the 5 s time base (Fig. 1b). Previous studies in which ONA was successfully applied implemented a 1 s time base (Hagler et al., 2011; Van den Bossche et al. 2015). After postprocessing with LPR and CMA, the microenvironmental characters retained more detailed information of the eBC concentration. Further comparison of their negative values revealed that remaining negative values comprised 28.1 % for LPR and 22.9 % for CMA after postprocessing.

In the 10 s interval time base, the negative values were not found after ONA processing, suggesting that a reasonable smoothing effect is obtained at low black carbon concentration. The microenvironmental character presented strong changes against the raw data, remaining less detailed information of air pollution. After postprocessing with LPR and CMA, the microenvironmental characters revealed more detailed information of air pollution, with 30.2 % of negative values for LPR and 25.3 % for CMA. In the 30 s interval time base, the negative values comprised 0 % of the post-processed data for ONA, 25.5 % for LPR, and 22.4 % for CMA. The 30 s interval dataset presented the lowest proportion of negative values before and after postprocessing, due to the longer interval times of sampling. However, the longer 30 s measurement period results in more distance covered during each measurement, given the mobile nature of the sampling device. Thus, 30 s black carbon measurements may be too long to detect local concentration peaks in urban contexts that supported in other study (Kerckhoffs et al., 2016).

The ONA algorithm showed a strong ability to extract negative values. As a result, the ONA-treated data may present bias that obscure nuanced microenvironmental trends (Fig. 1b). Interestingly, LPR and CMA postprocessing are capable of decreasing negative values while retaining microenvironmental trends. Both methods are promising for the analysis of spatiotemporal changes in pollutant concentrations with sensitivity to local sources. Previous studies have shown that the spatiotemporal variability of black carbon is highly heterogeneous (Liu et al., 2019; Liu et al., 2021); the ability to capture spatiotemporal variability of microenvironments is critical for assessing differential exposures among populations.

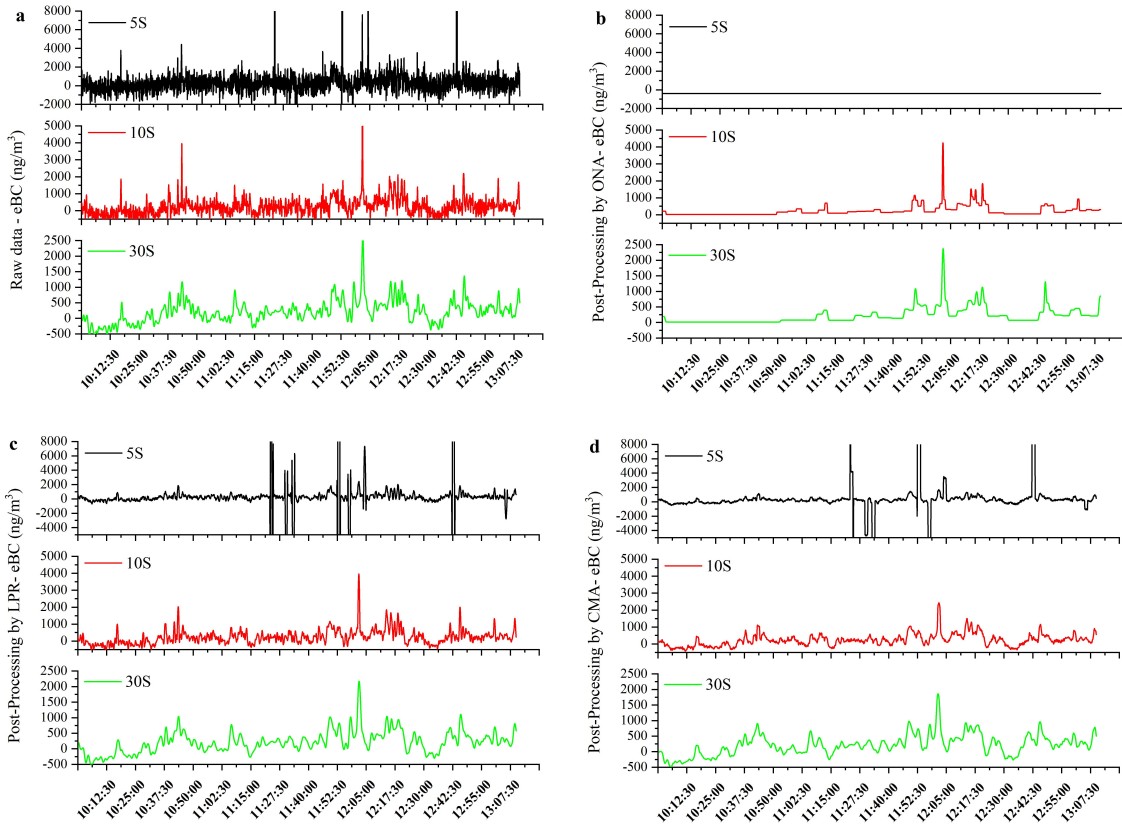

**Figure 1** The temporal fluctuations of the black carbon levels measured with the MA200 at sampling time bases of 5 s, 10 s, and 30 s during a typical sampling period (about 190 min), **(a)**, raw data without noise reduction, **(b)**, data treated with optimized noise reduction averaging, **(c)**, data treated with local polynomial regression, and **(d)**, data treated with centered moving average. The analysis was carried out on data streams from three MA200s all collected during a single sampling run (Measurements 5, 6 and 7).

**Table 2** The proportion of negative values and average noise reduction under the different postprocessing methods (values are shown as (%), NV [%]: Proportion of negative values remained, NR [%]: Average noise reduction. -, no data, measurements 1-10).

| Interval time | Factor | RAW | ONA | LPR | CMA |
|---|---|---|---|---|---|
| 5 s | NV | 42.1 | - | 28.1 | 22.9 |
| | NR | 0 | 100 | 72.0 | 87.4 |
| 10 s | NV | 37.6 | 0 | 30.2 | 25.3 |
| | NR | 0 | 5.54 | 22.3 | 47.7 |
| 30 s | NV | 30.5 | 0 | 25.5 | 22.4 |
| | NR | 0 | 0.62 | 6.24 | 39.1 |

**3.2 Reduction and number of peak-samples after postprocessing methods**

The processing of peak-sample is a pivotal evaluation index for the measurement of time-averaged roadside air quality. Passing vehicles, for example, may bias estimates of typical local concentrations due to their contribution to the dataset of peak concentrations that may substantially related to arithmetic averages. Therefore, after noise reduction, we compare the reduction values and the retained number of peak-samples to further evaluate the postprocessing methods.

In the interval time 5 s, the average reduction of peak-samples for the LPR and CMA algorithms was 72.0 % and 87.4 %, respectively (as discussed above, the ONA method could not be used). In this interval time, the reduction of peak-samples was relatively high, indicating that when monitoring black carbon at low concentrations and high sample frequencies, the drastic noise may occur in the raw data, and the higher noise reduction may affect the actual values. Therefore, the suitable interval time should be considered when monitoring low eBC concentrations. In the interval time 10 s, the average reduction of peak-samples for the CMA (47.7 %) is higher than ONA (5.54 %) and LPR (22.7 %). In the interval time 30 s, CMA presented the greatest average reduction of peak-samples (39.1 %) compared to ONA (6.24 %) and LPR (0.62 %) (Table 2, Fig. S4b). The retention of peak-samples remaining after postprocessing was also assessed using the COV method (Measurements 1-10). The result showed that all three algorithms retained all peak-samples before and after postprocessing. In this regard, CMA retained all peak samples despite the highest reduction in their magnitude. Therefore, CMA highlights microenvironmental trends while preserving the identity of peak-samples, facilitating the identification of local pollution sources, and may thus be a better postprocessing method than ONA or LPR (Table 2, Fig. S4b).

To further characterize the distribution of peak-sample concentration under CMA, we performed an intensive graphical analysis on a single data stream (Measurement 4; Fig. 2). As shown in Figure 2, eBC values along the main roads and intersections were higher than other locations, presumably due in large part to stop-and-go traffic and cars in close proximity to the mobile monitor (Fig. 2). It can be seen from Figure 2a that the peak-samples of black carbon were mainly found in 4 locations, represented by red triangles. Vehicle counts and traffic in these locations vary depending on the time of measurement. The highest eBC values were repeatedly found in the streets with moderately high traffic volumes and dense coverage with relatively high buildings (street canyon situation), indicating that heterogeneity in air pollution concentrations in Augsburg and similar settings is largely caused by a combination of effects from traffic and topography (Buonanno et al., 2011). To determine whether peak-samples are due to local sources or instrumental artifacts, and to provide further evidence that traffic and topography effects are primary contributors to spatial heterogeneity in pollution concentrations, we compared the data measurements of the three collocated MA200 units during Measurements 5, 6, and 7. The results showed that there were no major differences in the hot spot areas (an indicator of considerable peak-samples) identified by the measurements of the three instruments (Fig. S5).

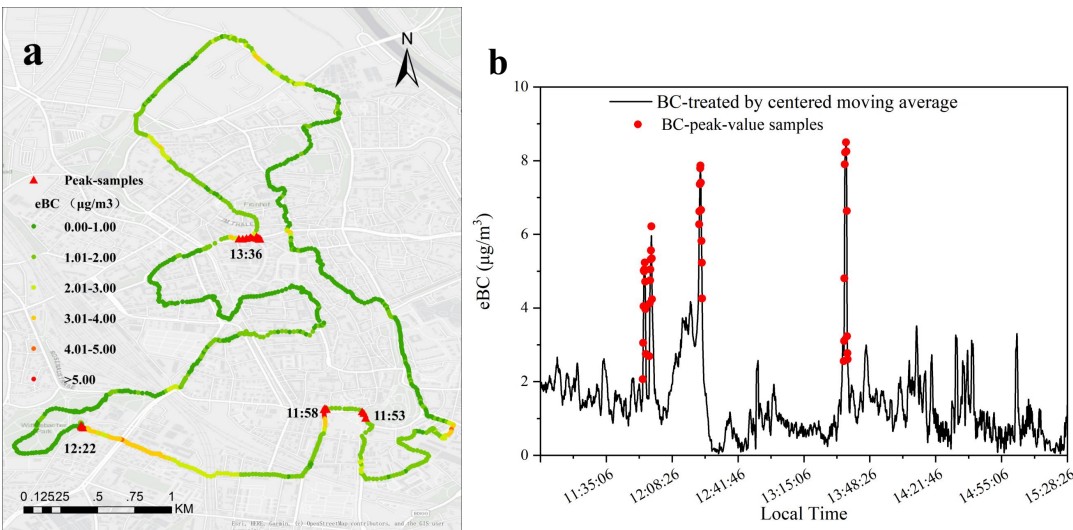

**Figure 2** Identification of the spatial **(a)** and temporal **(b)** distribution characteristics of black carbon peak-samples based on the coefficient of variation method (the analysis based on measurement 4), © OpenStreetMap contributors. Distributed under a Creative Commons BY-SA License.

It should be noted that a predecessor instrument to the MA200, the AE51, has demonstrated some sensitivity to mechanical shock during mobile measurements (Cai et al., 2013). Apte et al (2011) observed spurious, 1-3 second spikes of ± 200 - 2,000 µg/m$^3$ while monitoring black carbon in an auto rickshaw. When AethLabs took control of manufacturing the AE51, which was originally produced by Magee Scientific (Berkeley, CA, USA), instrument opto-electronics were redesigned to reduce such sensitivity (circa 2012). Researchers using redesigned AE51 demonstrated only a small effect on data. For example, Hankey (2014), using the same means of identifying such

spurious measurements as Apte et al (2011), observed that approximately 1-2 % of their data collected via bicycle trailer were attributed to spurious mechanical shock. Supporting this improvement, Cai et al (2013) found evidence of a substantial improvement in data quality related to vibration-related spikes after an equipment upgrade by AethLabs, which reflected the aforementioned improvements to opto-electronics. In addition, there were no major mechanical

shocks to or unique vibrational effects on the stroller and no major different of accelerometer data in the raw data, precluding these as potential con-founders on all 3 instruments.

**3.3 Comparison of background estimation and correction after noise reduction**

Local air pollution can be highly affected by long-range and regional transport. The timing and magnitude of such transport varies in space and time and is highly dependent upon the stochasticity of

meteorology. As a result, local background concentration changes may vary, affecting the comparability of measurements made at the same location at different times (Brantley et al., 2014). For this reason, reliable comparison of time-variable mobile measurements across a city (and thus reliable pinpointing of hotspots and pinpointing of key local sources) requires effective methods to estimate, isolate, and remove the effects of fluctuations in background concentration. Our analysis indicates that

the effectiveness of background correction is affected by the noise reduction method chosen during postprocessing.

After postprocessing, the data were evaluated using the TPRS method. We calculated the 5 min and 10 min background concentrations under different postprocessing approaches. As shown in Figures 3a and b, the background concentration after LPR processing has both the largest proportion of

negative values and the *most*-negative values (i.e. negative values of the greatest absolute magnitude), resulting in estimates of background-corrected concentrations that are greater than actual monitored concentrations. Background concentrations calculated after ONA and CMA postprocessing presented fewer and lower negative values than LPR, but were not convincingly different from each other. Therefore, to further compare the ONA and CMA algorithm, we also compared concentrations after

background correction (Fig. 3c and d). As shown in Figures 3c and d, when the concentration is lower than 1 $\mu g/m^3$ (black circle lines), the background-corrected results after the ONA processing are smoother than after CMA. This result dampens the signal of local pollutant sources, resulting in a lower utility of post-processed data.

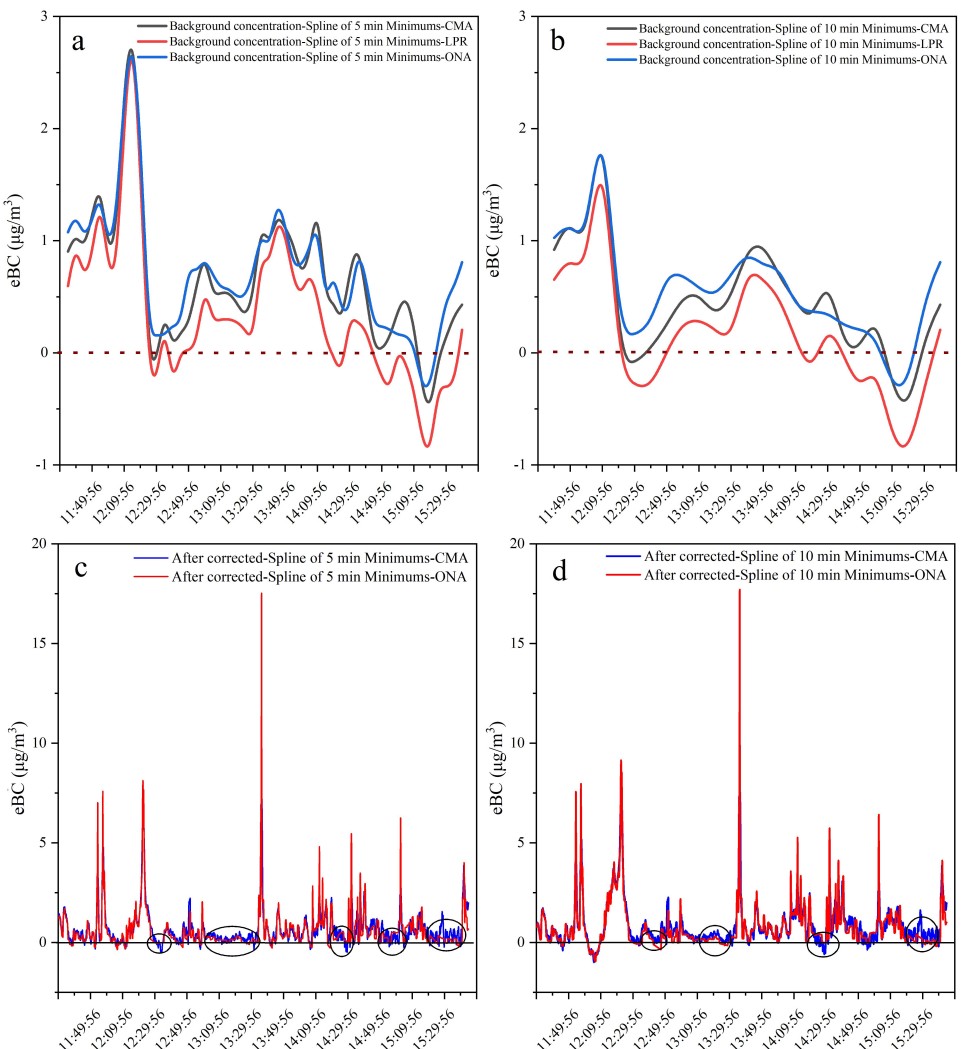

**Figure 3** Background concentration of black carbon under different time-series :**(a)**, spline of 5 min minimums, (b), spline of 10 min minimums; and background correction of black carbon under different time-series **(c)** spline of 5 min minimums; (d), spline of 10 min minimums. Analyses are based on Measurement 4.

In order to verify the CMA applicability and its advantages, this study further analyzed the eBC concentrations measured by a fixed background monitoring station at the University of Applied Sciences (UAS) (Fig. S6) (Cyrys et al., 2006). The background value under the 5 min window exhibits wave-like characteristics, and the fitting curve in the 10 min window is relatively smooth. However, the TPRS-based background value often does not fluctuate greatly over short periods, and the black carbon background value curve under the 5 min window does not conform to the "actual" urban background situation as estimated using the fixed-site monitor data, which are assumed to primarily represent the fluctuations in background concentrations. Moreover, by comparing the curve produced by the spline of 10 min minimums with the eBC background concentration (Background-UAS, Fig. S6), it can be found that the background correction method based on the time series can well characterize the

time-varying characteristics of background pollution in each experiment, suggesting that, of the two
options, 10 min showed the better window for fitting the background value curve of black carbon.

Under the TPRS method, the background concentration of eBC can be fitted at any sampling time. The
TPRS-estimated background contribution of the observed eBC concentration averaged 37.8 % of the
total measured concentration. However, when the contribution of background concentration to a single
measurement was examined, a large fluctuation (10.4 -71.3 %) was observed, which may be closely
related to sizeable changes in the meteorological conditions, traffic conditions along the road (and
overtime at the same point in the road), and urban street canyon effects in each measurement.
Therefore, based on the comparison of background correction, the CMA showed better applications for
estimating the background concentration and location source contribution.

### 3.4 Generalizability

To verify the generalizability of our assessment, we performed another three measurement runs in
Munich (Measurement 8, 9, 10). Raw data were post-processed for noise reduction using CMA (Fig.
S7). The results showed that the following method is equally applicable in a city like Munich as in our
study site in Augsburg, two cities that differ in location and environmental characteristics (e.g.,
population, economy, traffic density etc.). After treated by CMA, the peak-samples can be identified in
different interval times (Fig. S8), and the estimated background concentrations showed few negative
values (Fig. S9). Further research into the transferability of our results to a more diverse set of contexts
is still needed.

### 3.5 Practical implication

The MA200 is widely used to measure human exposure to black carbon and for mobile air quality
monitoring. In this study the MA200 were applied in mobile measurements in an urban area
(Augsburg), and the sensitivity of the final analysis to various data postprocessing methods was
investigated. In contrast to our findings, Hagler et al., (2011) suggested the use of ONA algorithm to
postprocess Aethalometer data from microAeth AE51, portable AE42, and rackmount AE21
aethalometers (Magee Scientific, Berkeley, CA, USA). In their analysis, ONA demonstrated a strong
noise reduction in all datasets and retained spatiotemporal variation. ONA also reduced the occurrence
of negative data values in low concentration sampling environments. However, for the microAeth®
series of black carbon monitoring instruments, our study showed that ONA leads to a considerable
dampening of spatiotemporal resolution in local black carbon signals at street level - an effect that is
lower under CMA postprocessing.

In addition, our analysis highlights that the selection of an appropriate data postprocessing method is
crucial to the proper assessment and interpretation of exposure-relevant microenvironmental
contributors to pollution concentrations in urban areas. This analysis is important when estimating
exposures that occur during transit, where spatiotemporal variability in pollution concentrations is vast,
like in commuter traffic (Snyder *et al.*, 2013). Due to the typically low-but-heterogeneous nature of

eBC concentrations in many areas like Augsburg, noisy measurement with the MA200 under high-frequency sampling may obscure actual trends in measured values. This study demonstrated that postprocessing MA200 data using CMA can reliably extract the actual signals from such noise and, alternatively, that postprocessing via ONA and LPR could be less reliable. Future researchers and agencies may find a distillation of our results in the form of the flow diagram in Scheme 1 useful in

determining how to reliably assess spatiotemporal variability of MA200 measurements for black carbon in different microenvironments.

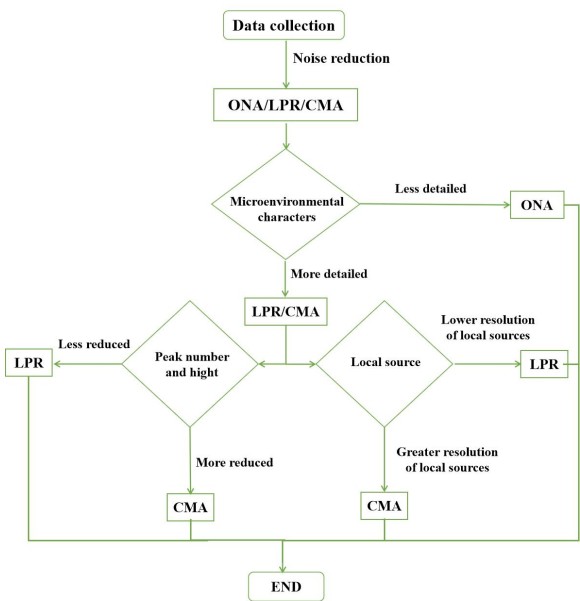

**Scheme 1** The proposed decision tree for mobile monitoring data from the microAeth® MA200.

**4 Conclusion**

A mobile monitoring campaign was conducted in the city center of Augsburg, Germany to determine a suitable noise reduction algorithm for the MA200 aethalometer. Our results showed that, at the interval time of 5 s, 10 s, and 30 s, CMA postprocessing effectively removed spurious negative concentrations without major bias and reliably highlighted effects from local sources, effectively increasing spatiotemporal resolution in mobile measurements. Evaluation of the effects of each method on

peak-sample reduction and the estimation of background concentrations further support the reliability of CMA algorithm. Further analysis is needed to understand how well these findings apply in different seasons; across different diurnal patterns; and in more-rural, more-urban, and non-German locations.

**Data availability**

The data is available upon request by contacting the first author of the paper.

## Author contribution

X.L: Data curation, Methodology, Software, Writing original draft. H.H: Methodology, Writing original draft. X.Z: Funding acquisition, Project administration. L. DH: Discussion, Writing review & editing. J.SK: Investigation, Supervision. J.B and G.L: Methodology. A. W and B.SH: Writing review & editing. RZ: Investigation.

## Competing Interest

The authors declare no conflict on interests.

**Acknowledgement**

We gratefully thank Erik Hopp (AethLabs, San Francisco, CA, USA) for the implementation and maintenance of the AethLabs data processing web tool.

**Financial support**

The work is funded by the Germany Federal Ministry of Transport and Digital Infrastructure (BMVI) as part of SmartAQnet (grant No.19F2003B), and by the Research Project of Ministry of Science and Technology of China (2019YFC0507800) and Support Project of High-level Teachers in Beijing Municipal Universities in the Period of 13th Five - year Plan (CIT&TCD201904037).

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
