# Peer review of "Analysis of mobile monitoring data from the microAeth® MA200 for measuring changes in black carbon on the roadside in Augsburg"

_Atmospheric Measurement Techniques, 2020_

## Referee Comment (RC1)

| GENERAL COMMENTS |
|---|

This manuscript provides an evaluation of three noise-reduction algorithms for the "raw" equivalent black carbon (eBC) mass concentration data of the new 5-wavelength microAethelometer model MA200. This has been submitted for the special issue: New developments in atmospheric limb measurements: instruments, methods, and science applications (AMT/ACP inter-journal SI).

Below are general comments on the manuscript as a whole:

**Appropriateness:**
1. *To the scope of the special issue (SI):* If this is not an error, it must be aptly justified why this is submitted in this SI. The SI call is specifically for "new developments in atmospheric limb measurements" focusing on the stratosphere. This manuscript has very little relevance to the scope of the SI and is an outlier among the other preprints included in this SI. I understand that the MA200 has been and may be used in vertical profiling. However, the dataset used here were from ground-based mobile measurements in an urban area. In my opinion, this manuscript does not belong in this SI.
2. *To the scope of AMT:* On the other hand, the manuscript does fall within the scope of AMT, in general. However, it lacks the detailed discussion on the technical aspects that is common with AMT publications, particularly if the presented "decision tree" is something the authors would like others to employ. **The entirety of the comments on this manuscript are based on this manuscript fitting the scope of AMT and not of the SI to which it was submitted to.**

**Scientific Significance and Quality:**

The idea behind this investigation is understandably important for some users of the MA200, particularly those who use the read out directly. However, the following issues were insufficiently addressed in the manuscript:
1. The motivation for noise-reduction, in general, was not sufficient. There is a part of the community who prefer the data as it is (since the instrumental noise cancels out when averaged), and focus instead on making sure the measurements are set-up correctly to prevent artificial peaks in the data (Cai et al., 2013; Alas et al., 2019).
2. It was mentioned in the text that the MA200 has an "on-board signal-processing that reduces the noise" of the MA200 raw data. Note that upon going through the user manual and quickly searching the MA200 website, this internal post-processing of the raw data is not mentioned (please, correct me if I'm wrong). This is an extremely important point that users need to know prior to using this instrument. This would mean that the output of the instrument is not "raw" anymore and, in the context of this study, the data has been "smoothed" out twice with the treatment of the noise-reduction algorithms being evaluated.
3. The measurement strategy was not explained in detail. Particularly, what measures were done to take into consideration the sensitivities of the MA200?
4. The results were merely presented. Deeper discussion on why the algorithms performed as they did is needed.
5. The broader significance of this study and how it relates to existing literature were not discussed.

**Presentation Quality:**

There is much to be improved with the writing of this manuscript. The main points are often times hidden in a mix of redundancy, jargon, lack of proper sentence/paragraph structures, and poor grammar. This provides so much hurdles for the readers in understanding the thought process of the author(s). Important aspects such as the criteria for evaluating the noise-reduction methods were often vague and leaves so much to interpretation or misinterpretation of the reader. Only my personal experience with the microAethalometer and mobile measurements allowed me to extract the information the author wants to give, and even then, it was with such difficulty.

The figures and tables, as well as their corresponding captions were not informative enough for the reader to understand them even after reading the manuscript (much less without). The figures and tables must be intuitive. Use of informative legends would improve the figures significantly. In addition, the parameters used to evaluate these noise-reduction algorithms were not defined prior to presentation of results. For instance, "noise reduction effect" and "negative decline rate" were not defined prior to showing up in Table 2 and how they were calculated was also missing.

*Some technical comments:*

- ➢ Define terms before abbreviating/ using initials (BC).
- ➢ Please consider using eBC (Petzold et al., 2013) consistently. It was anyway introduced in the text.
- ➢ Use initialisms throughout the manuscript.
- ➢ Be consistent with terminologies used.
- ➢ Please use "MA200" throughout the whole manuscript instead of switching from MA200 to "sample monitor" or "sampling equipment" every now and then.
- ➢ Use complete sentences in figure captions.
- ➢ The texts (titles, labels, legends) in the figures are not all the same size.

I strongly suggest major revisions in the writing with a native English speaker contributing on and reviewing the manuscript prior to re-submission.

| SPECIFIC COMMENTS | |
|---|---|
| Line # | Comments |
| | *texts in red are suggested changes in the manuscript to improve readability. |
| MAIN MANUSCRIPT | |
| ABSTRACT | |
| 32 | "Our results showed CMA to be a good prospect…" |
| 33 | This line is a little unclear. The readability may be improved. Here's a suggestion for this sentence:

"Based on the interval times used here, our results showed CMA to be a suitable algorithm to reduce the noise of raw BC mass concentration data based on the decrease of negative values and the retention of details attributable to microenvironmental changes." |
| 34 | Did you mean here "highest reduction OF peak values"? |
| 35-36 | "Furthermore, after background correction, the CMA results retained more detailed microenvironmental changes in pollutants than other methods." |
| 38-39 | "These findings provide new insights on the suitable noise reduction approach for mobile monitoring data obtained from portable BC instruments." |
| INTRODUCTION | |
| General Comments | • The jump in topics from line 48 to 49 is a bit big. I suggest to introduce at first the relevance of BC particles in air quality through its health effects. |

| | |
|---|---|
| | Then you continue with line 49. This will improve the motivation of your study. |
| | • Line 52: is it really the goal to propose a monitoring method or a method to analyze data from mobile monitoring? |
| | • It would not hurt to already introduce here the ONA method by Hagler et al. and to motivate why it is necessary to explore other means of reducing noise from the eBC datasets from portable absorption photometers. |
| | • There is insufficient motivation on the purpose/advantages of noise-reduction. |
| | • Explicitly mention that the noise-reduction algorithms evaluated here are those that are readily available in the Aethlabs Dashboard. There are members of the community who log AE51 or MA200 data independently and do not use the Dashboard. In any case, it must be justified, why the LPR and CMA are options in the Dashboard as noise-reduction algorithms. |
| 43 | "Black carbon particles with size ranging from …" |
| 67-68 | "…and simply removing negative values may introduce biases in the dataset." |
| | In this paragraph, it would be beneficial to inform the reader that these negative values are part of the instrumental noise, before you introduce noise reduction. |
| 70-72 | "Moreover, high-time resolution measurements of air quality at roadside are susceptible to single events (e.g. occasional passing of heavy-duty diesel vehicles or cigarette smoke) that may not be representative of the street in study. This may result in over estimation of eBC levels when averaged over time/space as they introduce peaks in the dataset." |
| 72-74 | "In addition, when the sampling equipment traverses from highly-polluted area to a low-polluted one, such as a park, the instrument produces strong negative peaks that is due to the measurement principle of the instrument and the strength of the pollution gradient between microenvironments." |
| 75 | "Therefore, the noise reduction method should also be evaluated based on the retention of actual peak concentrations and number of peak samples that are related to identifiable sources of pollution." |
| 78 | Remove "including" |
| 84 | The rationale of doing background correction in relation to the performance of noise reduction methods is not clear to me. Could you further elaborate on the statement that background concentration could affect the noise of the instrument? |
| 87- | This study evaluated three methods for reducing the noise from the raw BC dataset obtained using the MA200 in mobile measurements along a trafficked street in an urban area. The methods investigated are ONA (please cite here Hagler et al.,), LPR, and CMA. |
| 89-92 | "From these methods, the best noise reduction approach was selected by analyzing the post-processed results based on the following criteria: (1) relative number of negative values left; (2) retention of detailed information attributed to microenvironmental changes; (3) reduction of artificial peak values (is this correct?); and (4) retention of detailed information on microenvironmental changes after background correction (is this correct?)." |
| | METHODS |
| General comments: | • Some studies cited in this section used the older model AE51. Either clarify in one sentence or two that the studies you are referring here pertain to those that used a variety of portable absorption photometers for various applications, or remove citations which used the AE51 and not the MA200. |

| | |
|---|---|
| | • What methods were taken to account for the sensitivities of the instrument? The AE51 is known to be highly sensitive to vibrations, and sudden changes in the environment like RH and temperature. It is highly likely that the MA200 would have similar sensitivities, as also shown by Düsing et al. (2019) when it comes to strong RH variability. There was no detailed description on how the instrument was handled during the mobile measurements. This is vital since one of the criteria of the study is the retention of signals due to microenvironmental changes after noise reduction and identification of "peak samples". It must be noted that the instrument may produce false peaks/signals (in either direction) as a result of vibration or sudden change in RH and T. This is even more significant given that very few or even data from only a single mobile measurement round was used in the analysis. Taking these sensitivities into account in the measurement itself, would strengthen arguments made on the retention of "peak signals/samples".
 • You have to mention at some point that you only used data from one wavelength of the MA200.
 • Are the data already compensated? Was the internal DualSpot loading compensation operational during the measurements?
 • In which part of the analysis did you use Measurements 1-3? |
| 100 | "In mobile monitoring, the MA200 can be used to estimate personal exposure and quantify eBC mass concentrations in different microenvironments." |
| 108-109 | "In order to reduce the noise of the data obtained with high time resolution, smoothing algorithms can be used." |
| 112 | "…this study analyzed BC data collected from…" |
| 114 | • Why is it necessary to do further noise-reduction when there is already an on-board signal-processing? What is the principle behind the on-board signal processing and how does it differ from the methods investigated in this study? |
| 115 | Remove "microAeth®"
 Please provide some summary statistics of this comparisons. |
| 116-117 | Please clarify. Was the intercomparison between the MA200 and AE33 done during the walks (within one walk, the Ma200 stopped in the vicinity of the AE33 for short intercomparison)? How long were the intercomparisons? I understand that these results were presented in a previous publication, but summary statistics would aid readers. |
| 119 | Delete this: "To give intercomparison between the instruments…"
 You may start this sentence immediately at "To demonstrate the unit-to-unit comparability between the MA200 units, we performed intercomparisons at fixed monitoring stations and during collocated mobile measurements." |
| 138 | What does "To control for relative patterns in environmental exposure" mean? |
| 139 | "…the mobile measurements were carried out on the right side of the road simulating people's common habits…"" |
| 142 | I suggest either removing "air" (as it is vague) or "exposure" in this sentence. |
| 144 | Remove "with" after 4 h. |
| 149 | COV and TPRS are not yet defined prior to this. |
| 158-165 | This section could improve to briefly describe HOW the ONA reduce noise in microaethalometer data. How are the three parameters used to do this noise reduction? I believe this could greatly help readers in understanding Fig. S2 and, of course, the following analyses. LPR and CMA were aptly described in the following subsections, it would be great to elaborate a bit on ONA, too. |

| | |
|---|---|
| 169-173 | Please briefly describe "smoothing number", as the determination of this "smoothing number" is similar for that of CMA. How did you arrive at the values 15, 7, and 3? |
| 182 | "… the number of remaining negative values was determined."

 Also, what "number" of remaining negative values would imply a "good" noise reduction method? Is it simply a comparison of the treated data and whichever has the least number of negative values gets the point? |
| 182-189 | This paragraph could be greatly improved. I find the structuring of the sentence hard to understand, as well as the looping the same idea. Please simplify this and improve the writing for better readability.
 I suggest starting again here with the "criteria" you have for selecting the best noise-reduction approach. I understand you already enumerated them in line 145, but it was within the brief list of the process of the investigation. |
| 207 | It is unclear to me how the background estimation and correction is related to the investigation of the noise-reduction approaches. |
| 217-220 | This paragraph is better suited after the description of the noise reduction approaches.

 As for the 3rd criteria, it would help to specify what would make a noise reduction method "good". Is it it's ability to remove or retain these peaks?

 The criteria in judging which method is "good" should be crystal clear. |
| RESULTS AND DISCUSSION | |
| General comments | 3.1 → please improve the structure of the sentences
 3.2 → please explicitly distinguish between "peak samples" and "peak values"; and then in line 190 you also have "peak-value sample". These are all quite confusing.

 How are the "proportions retained" calculated? For instance, in the 5-s data, 42.1% of the raw data were negative values. After post-processing, "negative values retained 33.3% for LPR and 26.1% for CMA". Are the 33.3% and 26.1% from the total amount of negative values or from the whole dataset?
 Please include in your methods how these numbers are calculated.

 3.4 → why is background correction not applied to the Munich dataset? As I understand, one of the criteria for choosing CMA was its robustness to background correction. |
| 230 | Please elaborate on the explanation. I find it quite insufficient, particularly, in the ONA paper of Hagler et al., 2011, they published results of applying ONA on 1-s data of SootGen, stove, and mobile monitoring. Van den Bossche et al. (2015) also used ONA on 1-s data from AE51 in field measurements. Is this an instrument issue? Or an algorithm issue? |
| 237 | I do not understand the last part of this sentence. I think, I know what you are trying to say, but it's not coming across to the reader clearly. |
| 240 | Please be cautious of using the term "significant" here, particularly, that the analyses are based on comparability of statistical analyses of the raw data. I suggest the term "strong" here in place of "significant". |
| 242 | This is not a complete sentence. |
| 243 | Change "mitigating" to "decreasing". |

| | |
|---|---|
| 240-245 | A deeper discussion on the differences of the 3 noise-reduction approaches could greatly improve this part. In essence, this part was merely a presentation of results which are already in Table 2. |
| Fig. 2 | The unit should be nanograms. |
| | Am I right to assume that Fig. 2 is just same as Fig. 1 but only with the 10 s time resolution? |
| | If so, I do not see any added value in having this figure separated. The point you made in lines 240-245 is already clear in Fig. 1. |
| 255 Table 2 | It is unclear for me how the "noise reduction effect" was calculated. Please include in the methods section how these numbers are calculated and defined, including the "negative decline rate". |
| 257 | In this section, is my understanding correct? |
| | You want to evaluate two things about the "peaks": |
| | 1. # of peaks left after noise-reduction |
| | 2. Magnitude of these peaks after noise-reduction |
| | Is this right? |
| 264 | How is the "reduction effect" calculated? |
| 263-269 | It was not apparent right away that these results are already in Table 2. This could be solved by adding more information in the Table caption. Again, please give more information as to how these numbers are calculated or defined. Also, include the mean values in the table and not just the range so the readers can connect the numbers in this paragraph to the table. |
| | Do these numbers mean that CMA reduces the magnitude of the peak values greater than the other two noise reduction approaches? If so, what is the main criteria here? Do you want a noise-reduction algorithm that retains the magnitude of these peaks? Do you have a threshold where you say the algorithm diminished the peaks "too much"? |
| | A bar graph comparing raw and processed data for all your parameters would help clarify these compared to Table 2 alone. |
| 273-274 | This sentence is not clear. Did you mean to say, that based only on the # of remaining "peak samples", CMA performed better than the other approaches? |
| 288-289 | I do not understand how CMA, which "greatly reduces" the peaks (magnitude and number) is helpful in identifying "hotspots", in a sense. For instance, if this peak that is related to a source happens a few moments before or after a lower (below the COV threshold) peak, and it is greatly reduced by the CMA method, wouldn't that further blur the impact of this single source? I believe, a better criterion is a noise-reduction method that **does not greatly** reduce the magnitude of these peaks, particularly for exposure studies where every real signal is important. |
| Fig. 3 | The statement that these "spatial peaks" (Fig. 3a) are due to traffic and street canyon configuration could be better justified with a map that has spatially averaged eBC mass concentrations along the route. This also would prove the quality of the collocated measurements of the three MA200 and assure the reader that the peaks are due to local sources and not an instrumental artifact. I mean, you already have the data (running with 3 MA200 at the same time). |
| | Please provide more information in the figure caption such as the measurement number, to inform the readers that this is data from one run only. |

|  | Please also improve Fig. 3a by adding time stamps in the map to help readers reconcile the spatial plot with the time series. |
|---|---|
| 295 | This sentence can be simplified for better readability. |
| 306 | What is "minus absolute value"? |
| Fig. 4 | It is unclear if the figure 4 a and b are background concentration or background-corrected data. Please specify in the figure caption. |
|  | What is "actual detection concentration"? |
|  | What are the those encircled in dash black lines mean? Are they values below 1ug/m3? If so, it would help to draw a zero-line, or magnify the scale such that the data around 0 ug/m3 would be more visible. |
|  | Improve figure caption. |
| 318 | Change "certify" to "verify". |
| CONCLUSIONS | |
| General comments | • The broader significance of this study should be explicitly mentioned here. |
| 353-355 | The first sentence is misleading. As I understand, it was not the goal of this study to "assess BC pollution", but to determine a suitable noise reduction algorithm for the new MA200. |
| 369 | "The data is available upon request by contacting the first author of the paper." |
| 375 | "The authors declare no conflict of interests." |
|  | |
| SUPPORTING INFORMATION | |
| Table S1 | Are these numbers mean or median of the 5040 data points? Either way, please indicate and provide range, either quantiles, minimum and maximum, or standard deviation. How long were the measurements? |
| Table S2 | Another new terminology: "peak values number" ➢ Why is there no information for measurement numbers 5 and 7, 8-10? |
| Fig. S1 | Did you use standard major axis regression here to account for the error on both axes? Technically, none of these instruments are "reference" instruments to merit the use of simple linear regression. |
| Fig. S2 | Improve figure caption, indicate that this is for ONA. |
| Fig. S3 | Indicate that this is from CMA treated data. |
| Fig. S4 | So, the measurements in Munich were not simultaneous like in Augsburg? The figure labels are too big. Why is there no analysis of the "peak values" and "peak samples" for the Munich dataset? As I understand, you were testing the applicability of the CMA method to a different dataset, but fail to run the entire series of tests which "proved" CMA to be the suitable method. |

References:

COMMENTS TO THE AUTHOR:
amt-2020-517

Alas, H.D.C., Weinhold, K., Costabile, F., Di Ianni, A., Müller, T., Pfeifer, S., Di Liberto, L., Turner, J.R. and Wiedensohler, A. (2019). Methodology for high quality mobile measurement with focus on black carbon and particle mass concentrations. *Atmospheric Measurement Techniques* 12: 15.

Cai, J., Yan, B., Kinney, P.L., Perzanowski, M.S., Jung, K.H., Li, T., Xiu, G., Zhang, D., Olivo, C., Ross, J., Miller, R.L. and Chillrud, S.N. (2013). Optimization approaches to ameliorate humidity and vibration related issues using the microaeth black carbon monitor for personal exposure measurement. *Aerosol Sci Tech* 47: 1196-1204.

Düsing, S., Wehner, B., Müller, T., Stöcker, A. and Wiedensohler, A. (2019). The effect of rapid relative humidity changes on fast filter-based aerosol-particle light-absorption measurements: Uncertainties and correction schemes. *Atmospheric Measurement Techniques* 12: 5879-5895.

Petzold, A., Ogren, J.A., Fiebig, M., Laj, P., Li, S.M., Baltensperger, U., Holzer-Popp, T., Kinne, S., Pappalardo, G., Sugimoto, N., Wehrli, C., Wiedensohler, A. and Zhang, X.Y. (2013). Recommendations for reporting "black carbon" measurements. *Atmos Chem Phys* 13: 8365-8379.

Van den Bossche, J., Peters, J., Verwaeren, J., Botteldooren, D., Theunis, J. and De Baets, B. (2015). Mobile monitoring for mapping spatial variation in urban air quality: Development and validation of a methodology based on an extensive dataset. *Atmos Environ* 105: 148-161.

---

## Referee Comment (RC2)

In this manuscript, several methods were used for comparing the effect of noise reduction and negative value mitigation of the data from the microAeth® MA200. This topic of this manuscript is very interesting for the atmospheric limb measurements. However, I not only find many problems in this manuscript, but the methods used in this manuscript have been widely applied in other researches. Now, it seems there is no innovative in your manuscript. Therefore, I recommend reconsideration of your manuscript following **MAJOR** revision.

**<General Comments>**

**Scientific Significance and Quality:**

This manuscript applies a series of data post‐processing steps to microAeth® MA200 time series data to compare different method based on (1) the relative number of negative values; (2) more detailed microenvironmental change information retained after noise reduction; (3) the reduction of the peak values and number of peak samples; (4) more detailed microenvironmental change retained after the background correction. These methods can be important to properly characterize pollutant concentration data from mobile monitoring and demonstrate good practice for such applications. However, the authors are recommended to systematically compare the method and demonstrate the impact of various approaches and parameter settings.

Black carbon is a key indicator in air monitoring. Accurate measurement of black carbon is of great practical significance for the optimization of air quality. This manuscript compared the performance of various methods (e.g., LPR, ONA, and CMA) in noise reduction and negative value mitigation. It has high application value for users of the instrument. However, there are some issues that require serious consideration by the author.

1. These noise reduction methods are very common. Has the author considered the latest method or developed a more applicable method by himself?

2. MA200 is just one of many instruments. How valuable is your research for readers who do not use this instrument?

3. In the section "Results and discussion", I think the discussion part is relatively weak.

**Presentation Quality:**

I have to say that the presentation of the manuscript is very poor. There are many long sentences in

the text, which brings great dyslexia to readers. There are also many unreasonable expressions in paragraph structure and grammar. I suggest the author invite a native English speaker to rewrite the manuscript.

Some technical comments:

1. Consider using the latest references. References in the past three years only account for less than 30% of all your references.

2. Describe a full name and then its abbreviation throughout the manuscript.

3. The form of the pictures in the article is relatively simple. I suggest that you carefully modify the titles of the figures and tables.

4. Throughout the manuscript, the citation format of the figures is inconsistent, e.g., "Figure 4", "Fig. 3". You should keep them in the same format.

<Specific comments>

Line 25-28: "Noise reduction and negative value mitigation were explored via different data processing methods (e.g., local polynomial regression (LPR), optimized noise reduction averaging (ONA), and centered moving average (CMA)) under different interval time (i.e., 5s, 10s, and 30s)".

Line 30: "after noise reduction" repeated.

Line 31-33: I suggest this sentence "Our results showed that CMA showed a good prospect to analyze the raw BC concentration data in terms of the interval time due to its proportions of negative values and the detail microenvironmental change." should be split into several short sentences.

Line 34-35: I don't know what you want to express, please explain.

Line 35: "after background correction" appeared here. It also appeared in line 31, please carefully optimize the structure of the paragraph.

Line 39: BC instruments? I only saw MA200 in your manuscript.

Section 1 Unfortunately, I did not see you have a more detailed summary of the previous research. You only introduced the importance of black carbon measurement and the instruments you used. You should reflect the current research progress and deficiencies in this field in this section. Meanwhile, I suggest you cite some latest references.

Line 53-54: When is the specific development and market investment time of MA200?

Line 59-60: such as fossil fuel (e.g., diesel), biomass, and tobacco combustion

Line 60: "The instruments" or "This instrument"? Please check!

Line 63-64: There is a huge jump.

Line 65-66: "This is due to the use of an incremental optical attenuation value (ATN) to calculate the BC value." is not a correct sentence.

Line 77-86: Two references are not enough to explain the research progress of these contents, please consider adding references.

Line 110: What is your motivation for mentioning AE15 in this manuscript?

Line 123: "significant" is not a colloquial vocabulary. Its appearance usually requires standardized calculations. How did you get this conclusion?

Section 2.2: These redundant words make it difficult for me to understand your true intentions. It would be better if you could provide a framework figure.

Line 138: What does "relative patterns in environmental exposures" stand for?

Section 2.4: The introduction of the three methods is not sufficient. Who are their developers? What research field is it used for? What are the advantages and disadvantages? Specific formula? Since the comparison of three methods is the highlight of this article? You should give full attention instead of spending text in unimportant places.

Line 159: "the optical attenuation (ATN)", Line 65: "an incremental optical attenuation value (ATN)". This is a very irregular expression. I hope you can determine a correct expression about "ATN". This will bring greater difficulties to readers.

Line 174: "2.4.3. CMA (centered moving average)" should be changed to "2.4.3 CMA (centered moving average)"

Line 185: How did you define "more detailed"?

Line 191: Do you think the research 7 years ago (Brantley et al., (2014)) is the latest paper? This expression is inaccurate. I hope you can refer to the latest article. And the citation format of the references is also wrong. The correct one should look like this: A recent paper by Brantley et al. (2014) compared several methods for detecting and eliminating peak-value samples in mobile air pollution measurements.

Line 200: This passage is an explanation of the above formula. It does not belong to an independent sentence. Therefore, the first letter should be lowercase. "Where" to "where".

Line 200-201: The line spacing does not match the full manuscript.

Section 3: The discussion part is not sufficient.

Section 3.1: The author did not perform a significance test after data processing. This is a huge flaw. In addition, there are problems with the structure of many sentences.

Line 223: I suggest changing "three" to "3". In an article, the number format should be consistent. Please check the numbers that appear in the full manuscript.

Line 226: "Figs. 1b, 1c and 1d" is only part of Figure 1. "Fig. 1b, 1c, and 1d" is a more accurate way of expression.

Line 249-251: "The analysis based on data from measurements 5, 6, and 7, that were one run with three MA200 measuring parallel." is not a complete sentence.

Section 3.2: The results of this section is very interesting.

Section 3.3: "Comparison of background estimation and correction after noise reduction methods" should be replaced by "Comparison of background estimation and correction after noise reduction".

Line 294-296: This sentence should be simplified.

Line 299-301: "However, after different noise reduction approaches, the background correction concentration is different, therefore, further evaluation on their background correction concentration was necessary for this study." should be replaced by "However, the background correction concentration is different via different noise reduction approaches. Therefore, further evaluation on their background correction concentration was necessary for this study."

Line 302: Delete the first "methods".

Line 302-303: The structure of this sentence is so improper.

Line 317-320: Change this sentence to several short sentences.

Line 363: Delete "the centered moving average".

It seems to me that the whole manuscript does not have a decent map of the study area. In addition, lots of number expressions, made me lost in the jungle of numbers. Discover the hidden meaning of the numbers as much as you can.

Supporting information: You need to ensure the relative consistency of the font size in these figures.

Table S1: You need to give more detailed information, e.g., maximum, minimum, median, observation period, etc.

Table S2: Why does the "Measurement number" jump?

Figure S3: In order to improve the visibility of the curve, you can consider reducing the thickness of the curve.

---

## Author Response (AR1)

**Response to reviewer #1**

Thank you very much for your consideration of our manuscript (amt-2020-517). We consider the comments from you very constructive, and would like to thank you for the fine effort. Accordingly, we have made careful modifications. The revised manuscript has been reorganized, proof-read by a language professional, and marked in blue. The following are our detailed responses to each comment.

Reviewer #1
COMMENTS TO THE AUTHOR:
amt-2020-517
GENERAL COMMENTS
This manuscript provides an evaluation of three noise-reduction algorithms for the "raw" equivalent black carbon (eBC) mass concentration data of the new 5-wavelength microAethelometer model MA200. This has been submitted for the special issue: New developments in atmospheric limb measurements: instruments, methods, and science applications (AMT/ACP inter-journal SI).
Below are general comments on the manuscript as a whole:
Appropriateness:
1. To the scope of the special issue (SI): If this is not an error, it must be aptly justified why this is submitted in this SI. The SI call is specifically for "new developments in atmospheric limb measurements" focusing on the stratosphere. This manuscript has very little relevance to the scope of the SI and is an outlier among the other preprints included in this SI. I understand that the MA200 has been and may be used in vertical profiling. However, the dataset used here were from ground-based mobile measurements in an urban area. In my opinion, this manuscript does not belong in this SI.

**Response**: Thank you for bringing this to our more immediate attention. We agree and will take efforts to move the manuscript to the regular edition of AMT.

2. To the scope of AMT: On the other hand, the manuscript does fall within the scope of AMT, in general. However, it lacks the detailed discussion on the technical aspects that is common with AMT publications, particularly if the presented "decision tree" is something the authors would like others to employ. The entirety of the comments on this manuscript are based on this manuscript fitting the scope of AMT and not of the SI to which it was submitted to.

**Response**: Thank you for your comments. We will take efforts to move the manuscript to the regular edition of AMT. We have also added more content to the paper surrounding technical discussion and justification for the novelty and utility of the material to the Introductory Discussion sections.

Scientific Significance and Quality:
The idea behind this investigation is understandably important for some users of the MA200, particularly those who use the read out directly. However, the following issues were insufficiently addressed in the manuscript:
1. The motivation for noise-reduction, in general, was not sufficient. There is a part of the

community who prefer the data as it is (since the instrumental noise cancels out when averaged), and focus instead on making sure the measurements are set-up correctly to prevent artificial peaks in the data (Cai et al., 2013; Alas et al., 2019).

**Response**: Thank you for your correction and for pointing out the lack of clarity regarding our interests in "removing" negative values. We understand that a part of the community prefer the data as it is. Our paper is intended to serve and inform members of the community interested in smoothing out noise to produce more highly temporally resolved, unbiased estimates of eBC concentrations. We strongly agree that noise within the data, including negative values, contain valid information and that arbitrary removal of negative values may be detrimental to a dataset. We have updated the text to more make this more clear (e.g. the 5[th]-to-last paragraph in the Introduction). Specifically, our motivation is that all measurements have noise, and in the case of the Aethalometer® method and thus with the microAeth®, some data points will be greater than the actual value and some will be lower than the actual value. To understand data at a temporal precision of the data collection (e.g. a 1 hz time-base) with greater likelihood of matching the "actual" value, unbiased noise-reduction methods must be applied.

We also understand that, in fixed black carbon monitoring, concentrations are often more stable and less noisy across time, and, as a result, noise reduction is less critical. However, when we perform mobile monitoring, due to the high heterogeneity of black carbon concentrations, noise in the data can (often) more meaningfully affect the interpretation of temporal and, thus, spatial variations in black carbon concentrations at higher sample time bases (e.g., 10 s). In the absence of noise reduction or by simply averaging the monitoring data over discrete time periods, it is difficult to accurately observe spatial pollutant properties and scale. For mobile monitoring, therefore, data noise reduction is considered necessary for the MA200 (and many of other air quality monitors).

2. It was mentioned in the text that the MA200 has an "on-board signal-processing that reduces the noise" of the MA200 raw data. Note that upon going through the user manual and quickly searching the MA200 website, this internal post-processing of the raw data is not mentioned (please, correct me if I'm wrong). This is an extremely important point that users need to know prior to using this instrument. This would mean that the output of the instrument is not "raw" anymore and, in the context of this study, the data has been "smoothed" out twice with the treatment of the noise-reduction algorithms being evaluated.

**Response**: Thank you for your concern. The raw black carbon concentration data are the data directly outputted by the instrument, and our analysis is only postprocessed once. To avoid misunderstanding, we deleted this sentence.

3. The measurement strategy was not explained in detail. Particularly, what measures were done to take into consideration the sensitivities of the MA200?

**Response**: Our main measurement strategy is how monitoring devices, noise, and the application of different data postprocessing methods affect the measurement of black carbon in different microenvironments, under different instrumentation interval times, and, specifically, for mobile monitoring. More detailed information has been added to the revised manuscript. Please refer to line 171-176 for confirmation.

4. The results were merely presented. Deeper discussion on why the algorithms performed as they did is needed.

**Response**: Thank you for your suggestion. We have provided a deeper discussion on the algorithms in the revised manuscript. Please refer to the result and discussion section for confirmation.

5. The broader significance of this study and how it relates to existing literature were not discussed.

**Response**: Thank you for your suggestion. We have updated to text to more directly address the significance of this study and how it relates to existing literature in the "3.5 practical implication" section (line 453-478), and have added more justification for the work in the introduction.

Presentation Quality:

There is much to be improved with the writing of this manuscript. The main points are often times hidden in a mix of redundancy, jargon, lack of proper sentence/paragraph structures, and poor grammar. This provides so much hurdles for the readers in understanding the thought process of the author(s). Important aspects such as the criteria for evaluating the noise-reduction methods were often vague and leaves so much to interpretation or misinterpretation of the reader. Only my personal experience with the microAethalometer and mobile measurements allowed me to extract the information the author wants to give, and even then, it was with such difficulty.

The figures and tables, as well as their corresponding captions were not informative enough for the reader to understand them even after reading the manuscript (much less without). The figures and tables must be intuitive. Use of informative legends would improve the figures significantly. In addition, the parameters used to evaluate these noise-reduction algorithms were not defined prior to presentation of results. For instance, "noise reduction effect" and "negative decline rate" were not defined prior to showing up in Table 2 and how they were calculated was also missing.

**Response**: Thank you for your suggestion. We have revised the manuscript, including text, some figures, and tables together with the captions following your comments and suggestions to improve the readability of our manuscript.

Some technical comments:

1. Define terms before abbreviating/ using initials (BC).

**Response**: Thank you for your correction. In order to improve the clarity of black carbon, we choose to not abbreviate this term in our revised manuscript. We have carefully checked the revised manuscript for other abbreviations and acronyms

2. Please consider using eBC (Petzold et al., 2013) consistently. It was anyway introduced in the text.

**Response**: Thank you for your correction. We have carefully checked and revised through our manuscript regarding your concern, including the following text in the paragraph: "In our study, the equivalent black carbon (eBC), the preferred term for describing black carbon assessed with mass absorption cross-section (MAC) facilitated optical absorption methods (Petzold et al 2013), was used when addressing quantitative values." Please refer to line 77-79 for further confirmation.

3. Use initialisms throughout the manuscript.

**Response**: Thank you for your correction. We have carefully checked and revised through our manuscript regarding your concern (e.g., CMA, ONA, LPR). We have highlighted all related initialisms term on blue color. Please refer to our revised manuscript for further confirmation.

4. Be consistent with terminologies used.

**Response**: Thank you for your correction. We have carefully checked and revised our manuscript regarding your concern. Please refer to our revised manuscript for further confirmation.

5. Please use "MA200" throughout the whole manuscript instead of switching from MA200 to "sample monitor" or "sampling equipment" every now and then.

**Response**: Thank you for your correction. We have modified and revised through our manuscript regarding your concern. Please refer to our revised manuscript for further confirmation.

6. Use complete sentences in figure captions.

**Response**: Thank you for your correction. We have modified and used complete sentences in all figure captions including supplementary file in our revised manuscript. Please refer to our revised manuscript for further confirmation.

7. The texts (titles, labels, legends) in the figures are not all the same size.

**Response**: Thank you for your correction. We have checked and revised all the texts (titles, labels, legends) in the figures with the same size. Please refer to our revised manuscript for further confirmation.

8. I strongly suggest major revisions in the writing with a native English speaker contributing on and reviewing the manuscript prior to re-submission.

**Response**: We gave our revised manuscript to a native English speaker to improve the writing quality and readability.

SPECIFIC COMMENTS
MAIN MANUSCRIPT
ABSTRACT
1. line 32 "Our results showed CMA to be a good prospect…"

**Response**: Thank you for your correction. We have modified it combined with your comment. Please refer to line 32-33 for further confirmation.

2. line 33 This line is a little unclear. The readability may be improved. Here's a suggestion for this sentence: "Based on the interval times used here, our results showed CMA to be a suitable algorithm to reduce the noise of raw BC mass concentration data based on the decrease of negative values and the retention of details attributable to microenvironmental changes."

**Response**: Thank you for your correction. We have modified it combined with your comment. Please refer to line 32-35 for further confirmation.

3. line 34 Did you mean here "highest reduction OF peak values"?

**Response**: Yes it is. Please refer to line 35 for further confirmation.

4. line 35-36 "Furthermore, after background correction, the CMA results retained more detailed microenvironmental changes in pollutants than other methods."

**Response**: Thank you for your correction. We have modified it combined with your comment. Please refer to line 35-37 for further confirmation.

5. line 38-39 "These findings provide new insights on the suitable noise reduction approach for mobile monitoring data obtained from portable BC instruments."

**Response**: Thank you for your correction. We have modified it following your comment. Please refer to line 37-40 for further confirmation.

INTRODUCTION

General Comments

1. The jump in topics from line 48 to 49 is a bit big. I suggest to introduce at first the relevance of BC particles in air quality through its health effects. Then you continue with line 49. This will improve the motivation of your study.

Line 43 "Black carbon particles with size ranging from …"

**Response**: Thank you for your correction. We have modified it following your comment. Please refer to line 43-50 for further confirmation.

2. Line 52: is it really the goal to propose a monitoring method or a method to analyze data from mobile monitoring? It would not hurt to already introduce here the ONA method by Hagler et al. and to motivate why it is necessary to explore other means of reducing noise from the eBC datasets from portable absorption photometers.

There is insufficient motivation on the purpose/advantages of noise reduction?.

Explicitly mention that the noise-reduction algorithms evaluated here are those that are readily available in the Aethlabs Dashboard. There are members of the community who log AE51 or MA200 data independently and do not use the Dashboard. In any case, it must be justified, why the LPR and CMA are options in the Dashboard as noise-reduction algorithms.

**Response**: Thank you for your correction. We have briefly described and confirmed regarding with the goal of our study "to propose a monitoring method and motivation of noise reduction". Following that, we briefly introduce the ONA method by Hagler et al. to explore other means of reducing noise from the eBC datasets from portable absorption photometers. And justification of the LPR and CMA are options in the Dashboard as noise-reduction algorithms was also mentioned. Taken together, please refer to line 54-66 for further confirmation.

3. Line 67-68 "…and simply removing negative values may introduce biases in the dataset." In this paragraph, it would be beneficial to inform the reader that these negative values are part of the instrumental noise, before you introduce noise reduction.

**Response**: Thank you for your suggestion. We have added related information that negative values are part of the instrumental noise before introduce noise reduction. Please refer to line 81-83 for

further confirmation.

4. Line 70-72 "Moreover, high-time resolution measurements of air quality at roadside are susceptible to single events (e.g. occasional passing of heavy-duty diesel vehicles or cigarette smoke) that may not be representative of the street in study. This may result in over estimation of eBC levels when averaged over time/space as they introduce peaks in the dataset."
Line 72-74 "In addition, when the sampling equipment traverses from highly-polluted area to a low-polluted one, such as a park, the instrument produces strong negative peaks that is due to the measurement principle of the instrument and the strength of the pollution gradient between microenvironments."
Line 75 "Therefore, the noise reduction method should also be evaluated based on the retention of actual peak concentrations and number of peak samples that are related to identifiable sources of pollution."
Line 78 Remove "including"
**Response**: Thank you for your correction. We have modified it combined with your comment. Please refer to line 86-95 for further confirmation.

5. Line 84 The rationale of doing background correction in relation to the performance of noise reduction methods is not clear to me. Could you further elaborate on the statement that background concentration could affect the noise of the instrument?
**Response:** The background concentration does not affect the instrument noise reduction, but rather the instrument noise reduction affects the true value of the background concentration. Therefore, noise reduction, which better reflects the background concentration, is one of the criteria for assessing noise reduction in this study.

6. Line 87 This study evaluated three methods for reducing the noise from the raw BC dataset obtained using the MA200 in mobile measurements along a trafficked street in an urban area. The methods investigated are ONA (please cite here Hagler et al.,), LPR, and CMA.
**Response**: Thank you for your correction. We have cited the related reference (Hagler et al., 2011) regarding ONA in the revised manuscript. Please refer to line 107 for further confirmation.

7. 89-92 "From these methods, the best noise reduction approach was selected by analyzing the post-processed results based on the following criteria:
(1) relative number of negative values left;
(2) retention of detailed information attributed to microenvironmental changes;
(3) reduction of artificial peak values (is this correct?); and
(4) retention of detailed information on microenvironmental changes after background correction (is this correct?)."
**Response**: Thank you for your correction. That is correct (3rd and 4th criteria). And we have made modification regarding the criteria on the sentence. Please refer to line 110-113 for further confirmation.

METHODS
General comments:

1. Some studies cited in this section used the older model AE51. Either clarify in one sentence or two that the studies you are referring here pertain to those that used a variety of portable absorption photometers for various applications, or remove citations which used the AE51 and not the MA200.

**Response**: Thank you for your correction. AE51 is a predecessor instrument to the MA200, and this instrument has demonstrated some sensitivity to mechanical shock during mobile measurements. Therefore, We mentioned the instrument AE51 as a reference for MA200.

COMMENTS TO THE AUTHOR:
1. What methods were taken to account for the sensitivities of the instrument? The AE51 is known to be highly sensitive to vibrations, and sudden changes in the environment like RH and temperature. It is highly likely that the MA200 would have similar sensitivities, as also shown by Düsing et al. (2019) when it comes to strong RH variability. There was no detailed description on how the instrument was handled during the mobile measurements. This is vital since one of the criteria of the study is the retention of signals due to microenvironmental changes after noise reduction and identification of "peak samples". It must be noted that the instrument may produce false peaks/signals (in either direction) as a result of vibration or sudden change in RH and T. This is even more significant given that very few or even data from only a single mobile measurement round was used in the analysis. Taking these sensitivities into account in the measurement itself, would strengthen arguments made on the retention of "peak signals/samples".

**Response:** Thank you for your concern and suggestion. Firstly, the MA200's predecessor, the AE51 instrument, has shown some sensitivity to mechanical shock, relative humidity and temperature during mobile measurements. For mechanical shock, we have placed the MA200 in a carrying bag and secured it to a trolley, thus preventing mechanical shock during mobile sampling.

For the sensitivity of the MA200 to relative humidity and temperature, the comparative measurements of the MA200 and the stationary Aethalometer (AE33, Magee Scientific, Berkeley, USA) showed good agreement between stationary measurements taken during each walk. Briefly, the AE33 was used to monitor black carbon at the same time as the MA200, following that, the AE33 was placed in a container (fixed monitoring site) while the MA200 was used outdoors (in a stroller) while walking alone with different relative humidity and temperature. This phenomenon had no effect on the accuracy of the black carbon concentration for either instrument (Pearson's r = 0.933, below figure). The brief description about mechanical shock, relative humidity and temperature has been added in the revised manuscript. Please refer to line 140-147 and 383-396 for further confirmation.

*Mechanical shock.* The following text has been added to section 3.2:

"It should be noted that a predecessor instrument to the MA200, the AE51, has demonstrated some sensitivity to mechanical shock during mobile measurements (Cai et al., 2013). Apte et al (2011) observed spurious, 1-3 second spikes of $\pm$ 200 - 2,000 $\mu g/m^3$ while monitoring black carbon in an auto rickshaw. When AethLabs took control of manufacturing the AE51, which was originally produced by Magee Scientific (Berkeley, CA, USA), instrument opto-electronics were redesigned to reduce such sensitivity (circa 2012). Researchers using redesigned AE51 demonstrated only a small effect on data. For example, Hankey (2014), using the same means of identifying such spurious measurements as Apte et al (2011), observed that approximately 1-2 % of their data collected via bicycle trailer were attributed to spurious mechanical shock. Supporting this

improvement, Cai et al (2013) found evidence of a substantial improvement in data quality related to vibration-related spikes after an equipment upgrade by AethLabs, which reflected the aforementioned improvements to opto-electronics. In addition, there were no major mechanical shocks to or unique vibrational effects on the stroller and no major different of accelerometer data in the raw data, precluding these as potential con-founders on all 3 instruments."

*RH and Temperature:* The following text has been added to section 2.1:

In addition, it is worth noting that when the AE33 was used for monitoring black carbon at the same time as the MA200, the AE33 was placed in container, while MA200 was used outdoor (in the stroller) during the individual walks, which may have different relative humidity and temperature. This phenomenon did not influence the consistency of eBC concentration measured with both instruments.

[Figure]

**Figure R1** . The comparative measurements of the MicroAeth MA 200 with AE33.

2. You have to mention at some point that you only used data from one wavelength of the MA200.

**Response:** Thank you for your correction. We have modified and revised through our manuscript regarding your concern. Please refer to line 119-121 for further confirmation.

3. Are the data already compensated? Was the internal DualSpot loading compensation operational during the measurements?

**Response:** Yes, the data are already compensated.

Yes, it was internal DualSpot loading compensation operational during the measurements

4. In which part of the analysis did you use Measurements 1-3?

**Response:** The measurements 1-3 was used for the average black carbon concentrations of raw data, ONA-processed, LPR-processed, and CMA-processed data (Table S2), negative value proportion and noise reduction (Table 2), and COV analysis. Please refer to line 281, 341 and 357 for further confirmation.

5. line 100 "In mobile monitoring, the MA200 can be used to estimate personal exposure and quantify eBC mass concentrations in different microenvironments."

**Response:** Thank you for your correction. We have modified and revised through our manuscript regarding your concern. Please refer to line 123-125 for further confirmation.

6. line 108-109 "In order to reduce the noise of the data obtained with high time resolution,

smoothing algorithms can be used."

**Response:** Thank you for your correction. We have modified and revised through our manuscript regarding your concern. Please refer to line 132-134 for further confirmation.

7. line 112 "…this study analyzed BC data collected from…"

**Response:** Thank you for your correction. We have modified and revised through our manuscript regarding your concern. Please refer to line 137 for further confirmation.

8. line 114 Why is it necessary to do further noise-reduction when there is already an on-board signal-processing? What is the principle behind the on-board signal processing and how does it differ from the methods investigated in this study?

**Response**: Thank you for your concern. on-board signal-processing in MA200 reduce the noise of the black carbon signals, not the black carbon concentration. So, on-board signal will not influence the black carbon concentration. And the black carbon concentration data acquired by the instrument is the raw data without post-processing. Therefore, our analysis was only smoothed once. To avoid misunderstanding, we deleted this sentence.

9. line 115 Remove "microAeth®", Please provide some summary statistics of this comparisons.

**Response:** Thank you for your correction. We have removed "microAeth®" and added statistics summary of this comparisons in our revised manuscript. Please refer to line 140 for further confirmation.

10. 116-117 Please clarify. Was the intercomparison between the MA200 and AE33 done during the walks (within one walk, the Ma200 stopped in the vicinity of the AE33 for short intercomparison)? How long were the intercomparisons? I understand that these results were presented in a previous publication, but summary statistics would aid readers.

**Response:** Thank you for your correction. Yes it was, the intercomparisons between the MA200 and AE33 was done before and after walks for about 30 min to 60 min. The statistics summary of this comparisons was added in our revised manuscript. Following that, we have modified and revised through our manuscript regarding your concern to make the reader easier following our manuscript. Please refer to line 138-143 for further confirmation.

11. line 119 Delete this: "To give intercomparison between the instruments…" You may start this sentence immediately at "To demonstrate the unit-to-unit comparability between the MA200 units, we performed intercomparisons at fixed monitoring stations and during collocated mobile measurements."

**Response:** Thank you for your correction. We have modified and revised through our manuscript regarding your concern. Please refer to line 145-149 for further confirmation.

12. line 138 What does "To control for relative patterns in environmental exposure" mean?

**Response:** We are sorry that this sentence is difficult for the reader to understand.. It means "To control the different land use types of microenvironment". In order to avoid misunderstanding of the reader, we have modified and revised through our manuscript. Please refer to line 164 for

further confirmation.

13. line 139 "…the mobile measurements were carried out on the right side of the road simulating people's common habits…"
**Response:** Thank you for your correction. We have modified and revised through our manuscript regarding your concern. Please refer to line 165-166 for further confirmation.

14. line 142 I suggest either removing "air" (as it is vague) or "exposure" in this sentence.
**Response:** Thank you for your correction. We have removed "air" in this sentence.

15. line 144 Remove "with" after 4 h.
**Response:** Thank you for your correction. We have removed "with" in this sentence.

16. line 149 COV and TPRS are not yet defined prior to this.
**Response:** Thank you for your correction. We have introduced the complete terms of COV and TPRS. Please refer to line 181-182 for further confirmation.

17. line 158-165 This section could improve to briefly describe HOW the ONA reduce noise in microaethalometer data. How are the three parameters used to do this noise reduction? I believe this could greatly help readers in understanding Fig. S2 and, of course, the following analyses. LPR and CMA were aptly described in the following subsections, it would be great to elaborate a bit on ONA, too.
**Response**: Thank you for your correction. We have briefly described how the ONA reduce noise in microaethalometer data and revised through our manuscript regarding your concern. Please refer to line 199-210 for further confirmation.

18. line 169-173 Please briefly describe "smoothing number", as the determination of this "smoothing number" is similar for that of CMA. How did you arrive at the values 15, 7, and 3?
**Response**: The smoothing number of points is the number of data that need to be calculated from the original data. Following that the "smoothing number" is similar for that of CMA.
The smoothing number was obtained as the following formula
smoothing number = distance/ speed / interval time
100 (m)/1.3 (m/s)/5s=15.38≈15
100 (m)/1.3 (m/s)/10s=7.69≈7
100 (m)/1.3 (m/s)/30s=2.56≈3

19. line 182 "… the number of remaining negative values was determined." Also, what "number" of remaining negative values would imply a "good" noise reduction method? Is it simply a comparison of the treated data and whichever has the least number of negative values gets the point?
**Response**: Thank you for your correction. The "number" of remaining negative values would imply a "good" noise reduction method, when the treated data has a few number of negative values.

20. Line 182-189 This paragraph could be greatly improved. I find the structuring of the sentence hard to understand, as well as the looping the same idea. Please simplify this and improve the writing for better readability. I suggest starting again here with the "criteria" you have for selecting the best noise-reduction approach. I understand you already enumerated them in line 145, but it was within the brief list of the process of the investigation.

**Response:** Thank you for your correction. We have restructured the sentence and revised our manuscript regarding your concern to make the reader easier following our manuscript. Please refer to line 190-195 our revised manuscript for further confirmation.

21. line 207 It is unclear to me how the background estimation and correction is related to the investigation of the noise-reduction approaches.

**Response:** The background estimation and correction does not affect the instrument noise reduction, but rather the instrument noise reduction affects the true value of the background estimation and correction. We performed background estimation and correction to confirm which post-processing method performs better based on their background value and microenvironment character. Therefore noise reduction, which better reflects the background estimation and correction, is one of the criteria for assessing treated data in this study.

We have restructured the sentence and revised our manuscript regarding your concern to make the reader easier following our manuscript. Please refer to line 263-267 our revised manuscript for further confirmation.

22. Line 217-220 This paragraph is better suited after the description of the noise reduction approaches. As for the 3rd criteria, it would help to specify what would make a noise reduction method "good". Is it it's ability to remove or retain these peaks? The criteria in judging which method is "good" should be crystal clear.

**Response:** Thank you for your correction. We have restructured the sentence and revised our manuscript regarding your concern to make the reader easier following our manuscript. Please refer to line 190-195.

For the 3rd criteria, after noise reduction, we compare the reduction values and the number of peak samples to further evaluate the noise reduction methods. Briefly, when the reduction of peak value is high, the treated data has a high peak noise reduction without removing the numbers of peak-samples. Therefore, the method with high reduction of peak value and retaining the number of peak-samples after postprocessing was selected as the best method.

We have restructured the sentence and revised our manuscript regarding your concern to make the reader easier following our manuscript. Please refer to line 254-261 our revised manuscript for further confirmation.

RESULTS AND DISCUSSION
General comments
1. 3.1 please improve the structure of the sentences

**Response**: Thank you for your correction. We have restructured the 3.1 section sentence and revised our manuscript regarding your concern to make the reader easier following our manuscript. Please refer to line 286-341 our revised manuscript for further confirmation.

2. 3.2 please explicitly distinguish between "peak samples" and "peak values"; and then in line 190 you also have "peak-value sample". These are all quite confusing.

How are the "proportions retained" calculated? For instance, in the 5-s data, 42.1% of the raw data were negative values. After post-processing, "negative values retained 33.3% for LPR and 26.1% for CMA". Are the 33.3% and 26.1% from the total amount of negative values or from the whole dataset? Please include in your methods how these numbers are calculated.

**Response**: Thank you for your correction. We have determined to use "peak-samples" in all the revised manuscript. We have added how are the "proportions retained" calculated in the main text. Please refer to line 235-236 our revised manuscript for further confirmation.

3.3.4 why is background correction not applied to the Munich dataset? As I understand, one of the criteria for choosing CMA was its robustness to background correction.

**Response**: Thank you for your correction. We have analyzed the background correction of the Munich dataset and added it in the main text. As a result shown, after treated by CMA, the background concentrations showed few numbers of negative proportion (Fig. S8), suggesting the CMA method could be applied for black carbon postprocessing in another city. Please refer to line 449-450 and Fig. S8 our revised manuscript for further confirmation.

4. line 230 Please elaborate on the explanation. I find it quite insufficient, particularly, in the ONA paper of Hagler et al., 2011, they published results of applying ONA on 1-s data of SootGen, stove, and mobile monitoring. Van den Bossche et al. (2015) also used ONA on 1-s data from AE51 in field measurements. Is this an instrument issue? Or an algorithm issue?

**Response**: Thank you for your correction. As mentioned in section 2.2, the eBC average concentration is not high enough for this analysis in the city center of Augsburg, Germany, (measured at 2.62 μg/m$^3$ in winter by Gu, (2012)) thus in lower concentration, the ATN is more sensitive to the high time resolution. We have briefly elaborated this part in the revised manuscript. Please refer to line 295-298 for further confirmation.

5. line 237 I do not understand the last part of this sentence. I think, I know what you are trying to say, but it's not coming across to the reader clearly.

**Response**:Thank you for your correction. We have restructured the sentence and revised our manuscript regarding your concern to make the reader easier following our manuscript. Please refer to line 320-323 for further confirmation.

6. line 240 Please be cautious of using the term "significant" here, particularly, that the analyses are based on comparability of statistical analyses of the raw data. I suggest the term "strong" here in place of "significant".

**Response**:Thank you for your correction. We have modified the related sentence. Please refer to line 324 for further confirmation.

7. line 242 This is not a complete sentence.

**Response**:Thank you for your correction. We have modified the related sentence. Please refer to line 325-327 for further confirmation.

8. line 243 Change "mitigating" to "decreasing".

**Response**:Thank you for your correction. We have modified the related sentence. Please refer to line 326 for further confirmation.

9. line 240-245 A deeper discussion on the differences of the 3 noise-reduction approaches could greatly improve this part. In essence, this part was merely a presentation of results which are already in Table 2.

**Response**: Thank you for your correction. We have added more discussion in this part to improve the presentation of results. Please refer to line 324-331 for further confirmation.

10. Fig. 2 The unit should be nanograms. Am I right to assume that Fig. 2 is just same as Fig. 1 but only with the 10 s time resolution? If so, I do not see any added value in having this figure separated. The point you made in lines 240-245 is already clear in Fig. 1.

**Response**: Thank you for your correction. Fig. 2 (original version) with interval time 10 s is a part of Fig. 1. Therefore, following your suggestion, we removed the Fig. 2.

11. line 255 Table 2 It is unclear for me how the "noise reduction effect" was calculated. Please include in the methods section how these numbers are calculated and defined, including the "negative decline rate".

**Response**: Thank you for your correction. We have briefly described how to calculate the proportion of negative values and the reduction value of peak-samples in the method section 2.5.1 and 2.5.2, respectively. We have not longer used "negative decline rate" in our revised manuscript. Please refer to line 227-236 and 254-261 for further confirmation.

12. line 257 In this section, is my understanding correct? You want to evaluate two things about the "peaks": 1. # of peaks left after noise-reduction; 2. Magnitude of these peaks after noise-reduction Is this right?

**Response**: Yes, you are right. The text has been revised to make it more comprehensive.

13. line 264 How is the "reduction effect" calculated?

**Response**: Thank you for your correction. We have briefly described how to calculate the "the reduction value of peak-samples" in the method section 2.5.2. Please refer to line 254-261 for further confirmation.

14. line 263-269 It was not apparent right away that these results are already in Table 2. This could be solved by adding more information in the Table caption. Again, please give more information as to how these numbers are calculated or defined. Also, include the mean values in the table and not just the range so the readers can connect the numbers in this paragraph to the table.
Do these numbers mean that CMA reduces the magnitude of the peak values greater than the other two noise reduction approaches? If so, what is the main criteria here? Do you want a noise-reduction algorithm that retains the magnitude of these peaks? Do you have a threshold where you say the algorithm diminished the peaks "too much"? A bar graph comparing raw and processed data for all your parameters would help clarify these compared to Table 2 alone.

**Response**: Thank you for your suggestion. We have modified the Table 2 (adding the mean values) and added more information in the Table caption. As mentioned before, we have added how the proportion of negative values and the reduction value of peak-sample are calculated in the method section 2.5.1 and 2.5.2, respectively.

Yes, these numbers mean that CMA reduced the magnitude of the peak values greater than the other two noise reduction approaches, but retained the all number of peak-samples. Therefore, the criteria is the method with high reduction of peak value and retaining the number of peak-samples after postprocessing was selected as the best method. Following that, there is no specific threshold for the magnitude of peak reduction. However, it is same with the criteria as mentioned before. Following your suggestion, we have added a bar graph comparing raw and processed data for the negative values proportion and average reduction value of peak-sample (Fig. S4).

15. line 273-274 This sentence is not clear. Did you mean to say, that based only on the # of remaining "peak samples", CMA performed better than the other approaches?

**Response**: Thank you for your concern. Comparing the three postprocessing methods, CMA retains all number of peak samples despite the highest reduction in their magnitude, which highlights other micro-environmental characters and is helpful to identify the actual peak-sample location and further identify the source of pollution. However, ONA has lowest reduction, but it may omit micro-environmental characters, while LPR has higher reduction than ONA, but it retained higher proportion of negative value. Therefore, CMA performed better than the other approaches. We have restructured and modified the sentence regarding your concern to make the reader easier following our manuscript. Please refer to line 358-362 for further confirmation.

16. line 288-289 I do not understand how CMA, which "greatly reduces" the peaks (magnitude and number) is helpful in identifying "hotspots", in a sense. For instance, if this peak that is related to a source happens a few moments before or after a lower (below the COV threshold) peak, and it is greatly reduced by the CMA method, wouldn't that further blur the impact of this single source? I believe, a better criterion is a noise-reduction method that does not greatly reduce the magnitude of these peaks, particularly for exposure studies where every real signal is important.

**Response**: Thank you for your concern. After reanalysis for paek-samples identification, the three postprocessing methods have retained all the number of peak samples, but they have different reduction pattern of peak-samples after postprocessing. In this regard, CMA retained all peak samples despite the highest reduction in their magnitude. Therefore, CMA highlights microenvironmental trends while preserving the identity of peak-samples, facilitating the identification of local pollution sources, and may thus be a better postprocessing method than ONA or LPR (Table 2, Fig. S4b). Moreover, after CMA postprocessing, the treated data did not blur the effect of a single source and was useful to identify more sources or hotspots of air pollution. In order to avoid misunderstanding this part, we have restructured the sentence and revised our manuscript

regarding your concern to make the reader easier following our manuscript. Please refer to line 358-362 for further confirmation.

17. Fig. 3 The statement that these "spatial peaks" (Fig. 3a) are due to traffic and street canyon configuration could be better justified with a map that has spatially averaged eBC mass concentrations along the route. This also would prove the quality of the collocated measurements of the three MA200 and assure the reader that the peaks are due to local sources and not an instrumental artifact. I mean, you already have the data (running with 3 MA200 at the same time). Please provide more information in the figure caption such as the measurement number, to inform the readers that this is data from one run only. Please also improve Fig. 3a by adding time stamps in the map to help readers reconcile the spatial plot with the time series.

**Response**: Thank you for your suggestion. We have analyzed the three different MA200 to prove the peaks are due to local sources and not an instrumental artifact. The results showed there were no major differences in the hot spot areas shown by the measurements of the 3 instruments (Fig. S4). it further justified that these peak-samples were due to traffic and street canyon configuration. Following that, we have briefly described this part in our revised manuscript. In addition, we have provided more information in the figure caption and added time stamps in the map (Fig. 2a, revised manuscript), to help readers reconcile the spatial plot with the time series. Please refer to Fig. 2 and our supporting information for further confirmation.

18. Line 295 This sentence can be simplified for better readability.

**Response**: Thank you for your correction. We have restructured the sentence and revised our manuscript regarding your concern to make the reader easier following our manuscript. Please refer to line 398-400 for further confirmation.

19. line 306 What is "minus absolute value"?

**Response**: minus absolute value refers to *most*-negative values (i.e. negative values of the greatest absolute magnitude). We have revised this term in our manuscript regarding your concern to make the reader easier following our manuscript. Please refer to line 410 for further confirmation.

20. Fig. 4 It is unclear if the figure 4 a and b are background concentration or background corrected data? Please specify in the figure caption. What is "actual detection concentration"? What are the those encircled in dash black lines mean? Are they values below 1ug/m$^3$? If so, it would help to draw a zero-line, or magnify the scale such that the data around 0 ug/m$^3$ would be more visible. Improve figure caption.

**Response**: Thank you for your suggestion. We have modified the Fig. 4 (Fig. 3 after revised) following your suggestion together with the caption to improve the readability of the figure. The "actual detection concentration" is the measured concentration of the black carbon. The dash black lines (black circle lines after revised) indicated the background-corrected results after the ONA processing that values below 1ug/m$^3$. In order to make the reader easier following our manuscript, we draw a zero-line in the Figure 3 c and d. Please refer to line 419-423 for further confirmation.

21. Line 318 Change "certify" to "verify".

**Response:** Thank you for your correction. We have modified and revised through our manuscript

regarding your concern. Please refer to line 424 for further confirmation.

CONCLUSIONS

General comments

1. The broader significance of this study should be explicitly mentioned here.

Line 353-355 The first sentence is misleading. As I understand, it was not the goal of this study to "assess BC pollution", but to determine a suitable noise reduction algorithm for the new MA200.

**Response:** Thank you for your correction. We have modified and revised through our manuscript regarding your concern. Please refer to line 480-481 our revised manuscript for further confirmation.

2. line 369 "The data is available upon request by contacting the first author of the paper."

**Response:** Thank you for your correction. We have modified and revised through our manuscript regarding your concern. Please refer to line 489 for further confirmation.

3. line 375 "The authors declare no conflict of interests."

**Response:** Thank you for your correction. We have modified and revised through our manuscript regarding your concern. Please refer to line 496 for further confirmation.

SUPPORTING INFORMATION

1. Table S1 Are these numbers mean or median of the 5040 data points? Either way, please indicate and provide range, either quantiles, minimum and maximum, or standard deviation. How long were the measurements?

**Response**: Table S1 are mean of the 5040 data points, the measurements were performed for 14 h. In our perspective, the standard deviation has very limited meaningfulness, because these data are ambient air measurements with a diurnal pattern, so that the standard deviation is very high, nevertheless, we have analyzed and shown in the following able.

**Table S1** Comparative measurements of different MA200 in the fixed monitoring station (unit: $ng/m^3$, total N=5040 for each MA200, each measurement 14 h).

| Measurements | 375 nm | 470 nm | 528 nm | 625 nm | 880 nm |
|---|---|---|---|---|---|
| MA200-0051 | 818±183 | 833±226 | 812±224 | 810±232 | 774±251 |
| MA200-0053 | 827±115 | 838±121 | 814±124 | 815±128 | 783±164 |
| MA200-0059 | 870±186 | 866±155 | 830±163 | 840±161 | 814±213 |
| MA200-0060 | 872±121 | 881±135 | 857±126 | 857±135 | 822±169 |
| MA200-0155 | 856±103 | 855±112 | 842±107 | 840±115 | 830±138 |
| MA200-0153 | 846±153 | 850±180 | 822±109 | 832±117 | 795±152 |
| MA200-0159 | 825±108 | 845±148 | 818±108 | 832±110 | 780±157 |
| Mean | 844.9±22.1 | 852.6±16.6 | 827.9±16.5 | 832.3±15.9 | 799.7±22.2 |

2. Table S2 Another new terminology: "peak values number" Why is there no information for measurement numbers 5 and 7, 8-10?

**Response:** Thank you for your correction. After reanalysis all of the raw data and all postprocessing data (measurements 1-10), the number of peak samples did not change before and

after postprocessing. Therefore, this table is no longer used in our revised manuscript. The detail information about it, please refer to line 356-357 for further confirmation.

3. Fig. S1 Did you use standard major axis regression here to account for the error on both axes? Technically, none of these instruments are "reference" instruments to merit the use of simple linear regression.

**Response:** Figure S1 (Fig. S2 after revised) is presented to demonstrate the unit-to-unit comparability between the MA200 units in the black carbon concentration during collocated mobile measurements. The results showed that there were no significant wavelength dependence between different instruments in different interval times. Therefore, in our perspective, the standard major axis regression and "reference" instruments are very limited meaningfulness.

4. Fig. S2 Improve figure caption, indicate that this is for ONA.

**Response:** Thank you for your correction. We have improved Figure S2 (Fig. S3 after revised) caption. Please refer to our revised supplementary for further confirmation.

5. Fig. S3 Indicate that this is from CMA treated data.

**Response:** Thank you for your correction. We have improved Figure S3 caption (Fig. S6 after revision). Please refer to our revised supplementary for further confirmation.

6. Fig. S4 So, the measurements in Munich were not simultaneous like in Augsburg? The figure labels are too big. Why is there no analysis of the "peak values" and "peak samples" for the Munich dataset? As I understand, you were testing the applicability of the CMA method to a different dataset, but fail to run the entire series of tests which "proved" CMA to be the suitable method.

**Response:** Thank you for your correction. We have analyzed the "peak samples" (Fig. S8) and background concentration (Fig. S9) for the Munich dataset. The result showed that after treated by CMA, the peak-samples can be identified in different interval time (Fig. S8, and the background concentrations showed few numbers of negative proportion (Fig. S9).

Please refer to line 449-451 and supplementary for further confirmation.

**Response to reviewer #2**

Thank you very much for your consideration of our manuscript (amt-2020-517). We consider the comments from you very constructive, and would like to thank you for the fine effort. Accordingly, we have made careful modifications. The revised manuscript has been reorganized, proof-read by a language professional, and marked in blue. The following are our detailed responses to each comment.

Reviewer #2
In this manuscript, several methods were used for comparing the effect of noise reduction and negative value mitigation of the data from the microAeth® MA200. This topic of this manuscript is very interesting for the atmospheric limb measurements. However, I not only find many

problems in this manuscript, but the methods used in this manuscript have been widely applied in other researches. Now, it seems there is no innovative in your manuscript. Therefore, I recommend reconsideration of your manuscript following MAJOR revision.

<General Comments>

Scientific Significance and Quality:

This manuscript applies a series of data post‑processing steps to microAeth® MA200 time series data to compare different method based on (1) the relative number of negative values; (2) more detailed microenvironmental change information retained after noise reduction; (3) the reduction of the peak values and number of peak samples; (4) more detailed microenvironmental change retained after the background correction. These methods can be important to properly characterize pollutant concentration data from mobile monitoring and demonstrate good practice for such applications. However, the authors are recommended to systematically compare the method and demonstrate the impact of various approaches and parameter settings.

Black carbon is a key indicator in air monitoring. Accurate measurement of black carbon is of great practical significance for the optimization of air quality. This manuscript compared the performance of various methods (e.g., LPR, ONA, and CMA) in noise reduction and negative value mitigation. It has high application value for users of the instrument. However, there are some issues that require serious consideration by the author.

1. These noise reduction methods are very common. Has the author considered the latest method or developed a more applicable method by himself?

**Response**: Thank you for your comment. The latest method developed for black carbon noise reduction referred to the ONA treated data by Hagler et al., 2011. They published results of applying ONA on 1-s data of SootGen, stove, and mobile monitoring. Van den Bossche et al. (2015) also used ONA on 1-s data from AE51 in field measurements. However, the instrumentation for monitoring black carbon used in this study was MA200, which was different with the previous studies (i.e., AE51). In this instrument, the LPR and CMA algorithms were introduced for noise reduction instead of ONA algorithm. Although these two algorithms are the common methods. But their application is still relatively sparse. Therefore, in this study we assessed and evaluated the postprocessing method for black carbon data obtained by MA200.

2. MA200 is just one of many instruments. How valuable is your research for readers who do not use this instrument?

**Response**: Thank you for your correction. Nowadays, the portable microAeth® MA200 (MA200) is widely applied for measuring black carbon in human exposure characterization and mobile air quality monitoring. However, the field lacks information about this instrument's performance under various settings. Therefore, it is really important to provide this research (evaluation the real-time performance of the MA200 in an urban area) for MA200 users. Following that, this research also provided the information about postprocessing methods to non-users of MA200, which may also have applicability for other instruments.

3. In the section "Results and discussion", I think the discussion part is relatively weak.

**Response**: Thank you for your suggestion. We have added a deeper discussion in the "Results and discussion" section. Please refer to our revised manuscript for further confirmation.

Presentation Quality:

I have to say that the presentation of the manuscript is very poor. There are many long sentences in the text, which brings great dyslexia to readers. There are also many unreasonable expressions in paragraph structure and grammar. I suggest the author invite a native English speaker to rewrite the manuscript.

**Response**: Thank you for your suggestion. We have given a check our revised manuscript to a native English speaker to improve the writing quality and readability.

Some technical comments:

1. Consider using the latest references. References in the past three years only account for less than 30% of all your references.

**Response**: Thank you for your suggestion. We have added some recent references in our revised manuscript. Please refer to "references" section in our revised manuscript for further confirmation.

2. Describe a full name and then its abbreviation throughout the manuscript.

**Response**: Thank you for your correction. We have again carefully double checked about it and revised through our manuscript regarding your concern. Please refer to our revised manuscript for further confirmation.

3. The form of the pictures in the article is relatively simple. I suggest that you carefully modify the titles of the figures and tables.

**Response**: Thank you for your correction. We have checked and revised all the texts (titles, labels, legends) in the figures and tables with the same size and their captions, including supplementary file in our revised manuscript. Please refer to our revised manuscript for further confirmation.

4. Throughout the manuscript, the citation format of the figures is inconsistent, e.g., "Figure 4", "Fig. 3". You should keep them in the same format.

**Response**: Thank you for your correction. We have again carefully double checked about it and keep them in the same format. Please refer to our revised manuscript for further confirmation.

<Specific comments>

1. Line 25-28: "Noise reduction and negative value mitigation were explored via different data processing methods (e.g., local polynomial regression (LPR), optimized noise reduction averaging (ONA), and centered moving average (CMA)) under different interval time (i.e., 5s, 10s, and 30s)".

**Response**: Thank you for your correction. We have modified it combined with your comment. Please refer to line 26-29 for further confirmation.

2. Line 30: "after noise reduction" repeated.

**Response**: Thank you for your correction. We have deleted it following your comment. Please refer to line 29 for further confirmation.

3. Line 31-33: I suggest this sentence "Our results showed that CMA showed a good prospect to

analyze the raw BC concentration data in terms of the interval time due to its proportions of negative values and the detail microenvironmental change." should be split into several short sentences.

**Response**: Thank you for your correction. We have modified it combined with your comment. Please refer to line 32-35 for further confirmation.

4. Line 34-35: I don't know what you want to express, please explain.

**Response**: Thank you for your correction. After noise reduction, we compare the reduction values and the number of peak samples to further evaluate the noise reduction methods. Briefly, when the reduction of peak value is high, the treated data has a high peak noise reduction without removing the numbers of peak-samples. Therefore, the method with high reduction of peak value and retaining the number of peak-samples after postprocessing was selected as the best method. We have modified and revised through our manuscript regarding your concern to make the reader easier following our manuscript. Please refer to line 32-35 for further confirmation.

5. Line 35: "after background correction" appeared here. It also appeared in line 31, please carefully optimize the structure of the paragraph.

**Response**: Thank you for your correction. We have modified it following your comment.

6. Line 39: BC instruments? I only saw MA200 in your manuscript.

**Response**: Thank you for your correction. We have modified it following your comment. Please refer to line 39 for further confirmation.

7. Section 1 Unfortunately, I did not see you have a more detailed summary of the previous research. You only introduced the importance of black carbon measurement and the instruments you used. You should reflect the current research progress and deficiencies in this field in this section. Meanwhile, I suggest you cite some latest references.

**Response**: Thank you for your suggestion. We have added more detailed summary of the previous research (Hegler et al., 2011; Van den Bossche et al., 2015) that reflected the current research progress and deficiencies in this field. Please refer to line 55-60 for further confirmation.

8. Line 53-54: When is the specific development and market investment time of MA200?

**Response:** In 2017 AethLabs released the MA Series of Black carbon monitors with 3 different products that build on microAeth® AE51 personal exposure monitor. The MA200 is a compact and highly integrated personal exposure monitor often used for health effects and mobile transportation studies.

9. Line 59-60: such as fossil fuel (e.g., diesel), biomass, and tobacco combustion.

**Response**: Thank you for your correction. We have modified it following your comment. Please refer to line 74 for further confirmation.

10. Line 60: "The instruments" or "This instrument"? Please check!

**Response**: Thank you for your correction. We have double checked and modified it. Please refer to line 75 for further confirmation.

11. Line 63-64: There is a huge jump.

**Response**: Thank you for your correction. We have double checked and slight modified it. Please refer to line 80-81 for further confirmation.

12. Line 65-66: "This is due to the use of an incremental optical attenuation value (ATN) to calculate the BC value." is not a correct sentence.

**Response**: Thank you for your correction. We have deleted this sentence and modified it. Please refer to line 81-83 for further confirmation.

13. Line 77-86: Two references are not enough to explain the research progress of these contents, please consider adding references.

**Response**: Thank you for your correction. We have added more references following your comment. Please refer to line 96-105 for further confirmation.

14. Line 110: What is your motivation for mentioning AE51 in this manuscript?

**Response**: Thank you for your correction. AE51 is a predecessor instrument to the MA200, and this instrument has demonstrated some sensitivity to mechanical shock during mobile measurements. Therefore, We mentioned the instrument AE51 as a reference for MA200.

15. Line 123: "significant" is not a colloquial vocabulary. Its appearance usually requires standardized calculations. How did you get this conclusion?

**Response**: Thank you for your correction. We have modified the "significant" term in the sentence. Please refer to line 147-149 for further confirmation.

16. Section 2.2: These redundant words make it difficult for me to understand your true intentions. It would be better if you could provide a framework figure.

**Response**: Thank you for your correction. We have modified the 2.2 section and added the map of sampling to make the reader better understand following our manuscript. Please refer to 2.2 section in our revised manuscript for further confirmation.

17. Line 138: What does "relative patterns in environmental exposures" stand for?

**Response:** We are sorry to make you difficult to understand this sentence. It means "To control the different land use types of microenvironment". In order to avoid misunderstanding of the reader, we have modified and revised through our manuscript. Please refer to line 164 for further confirmation.

18. Section 2.4: The introduction of the three methods is not sufficient. Who are their developers? What research field is it used for? What are the advantages and disadvantages? Specific formula? Since the comparison of three methods is the highlight of this article? You should give full attention instead of spending text in unimportant places.

**Response**: Thank you for your correction. We have modified it following your comment. Please refer to section 2.4 for further confirmation.

19. Line 159: "the optical attenuation (ATN)", Line 65: "an incremental optical attenuation value (ATN)". This is a very irregular expression. I hope you can determine a correct expression about "ATN". This will bring greater difficulties to readers.

**Response**: Thank you for your correction. We have modified it following your comment. Please refer to line 71 for further confirmation.

20. Line 174: "2.4.3. CMA (centered moving average)" should be changed to "2.4.3 CMA (centered moving average)"

**Response**: Thank you for your correction. We have modified it following your comment. Please refer to line 220 for further confirmation.

21. Line 185: How did you define "more detailed"?

**Response**: We defined "more detailed" to microenvironmental characters information.

22. Line 191: Do you think the research 7 years ago (Brantley et al., (2014)) is the latest paper? This expression is inaccurate. I hope you can refer to the latest article. And the citation format of the references is also wrong. The correct one should look like this: A recent paper by Brantley et al. (2014) compared several methods for detecting and eliminating peak-value samples in mobile air pollution measurements.

**Response**: Thank you for your correction. Brantley et al., (2014) is most suitable reference according to identification of peak-samples in mobile air pollution measurements, therefore, we cited this reference in this part. We have modified it following your comment. Please refer to line 238 for further confirmation.

23. Line 200: This passage is an explanation of the above formula. It does not belong to an independent sentence. Therefore, the first letter should be lowercase. "Where" to "where".

**Response**: Thank you for your correction. We have modified it following your comment. Please refer to line 248 for further confirmation.

24. Line 200-201: The line spacing does not match the full manuscript.

**Response**: Thank you for your correction. We have modified it following your comment. Please refer to line 246 for further confirmation.

25. Section 3: The discussion part is not sufficient.

**Response**: Thank you for your suggestion. We have added a deeper discussion in the "Results and discussion" section. Please refer to our revised manuscript for further confirmation.

26. Section 3.1: The author did not perform a significance test after data processing. This is a huge flaw. In addition, there are problems with the structure of many sentences.

**Response**: We are sorry to make you misunderstanding. We have changed the related sentence in our revised manuscript to avoid misunderstanding of the readers and we have restructured the section 3.1 to improve the readability of the manuscript.

27. Line 223: I suggest changing "three" to "3". In an article, the number format should be

consistent. Please check the numbers that appear in the full manuscript.

**Response**: Thank you for your correction. We have modified it following your comment. Please refer to our revised manuscript for further confirmation.

28. Line 226: "Figs. 1b, 1c and 1d" is only part of Figure 1. "Fig. 1b, 1c, and 1d" is a more accurate way of expression.

**Response**: Thank you for your correction. We have modified it following your comment. Please refer to line 288 for further confirmation.

29. Line 249-251: "The analysis based on data from measurements 5, 6, and 7, that were one run with three MA200 measuring parallel." is not a complete sentence.

**Response**: Thank you for your correction. We have modified it following your comment. Please refer to line 336-338 for further confirmation.

30. Section 3.2: The results of this section is very interesting.

**Response**: Thank you for your positive comment.

31. Section 3.3: "Comparison of background estimation and correction after noise reduction methods" should be replaced by "Comparison of background estimation and correction after noise reduction".

**Response**: Thank you for your correction. We have modified it combined with your comment. Please refer to line 397 for further confirmation.

32. Line 294-296: This sentence should be simplified.

**Response**: Thank you for your correction. We have modified it combined with your comment. Please refer to line 398-400 for further confirmation.

33. Line 299-301: "However, after different noise reduction approaches, the background correction concentration is different, therefore, further evaluation on their background correction concentration was necessary for this study." should be replaced by "However, the background correction concentration is different via different noise reduction approaches. Therefore, further evaluation on their background correction concentration was necessary for this study."

**Response**: Thank you for your correction. We have modified it combined with your comment. Please refer to line 401-406 for further confirmation.

34. Line 302: Delete the first "methods".

**Response**: Thank you for your correction. We have deleted it following your comment.

35. Line 302-303: The structure of this sentence is so improper.

**Response**: Thank you for your correction. We have modified it following your comment. Please refer to line 407-408 for further confirmation.

35. Line 317-320: Change this sentence to several short sentences.

**Response**: Thank you for your correction. We have modified it following your comment. Please refer to line 424-426 for further confirmation.

36. Line 363: Delete "the centered moving average".

**Response**: Thank you for your correction. We have deleted it following your comment.

37. It seems to me that the whole manuscript does not have a decent map of the study area. In addition, lots of number expressions, made me lost in the jungle of numbers. Discover the hidden meaning of the numbers as much as you can.

**Response**: We are sorry inconvenient. Following your concern, we have added one map to show the study area **(Figure S1)**. In addition, we have added some figures to reduce a lot of number expressions and make the reader easier to follow our manuscript. For example, we added **Figure S4** to represent the negative values proportion and average noise reduction of this study. More than that we have restructured the whole manuscript to improve its readability. Please refer to support information for further confirmation.

38. Supporting information: You need to ensure the relative consistency of the font size in these figures.

**Response**: Thank you for your correction. We have checked and revised all the texts (titles, labels, legends) in the figures with the same size. Please refer to our revised manuscript for further confirmation.

39. Table S1: You need to give more detailed information, e.g., maximum, minimum, median, observation period, etc.

**Response**: Table S1 are mean of the 5040 data points, the measurements were performed for 14 h. In our perspective, the maximum, minimum, and median have very limited meaningfulness, because these data are ambient air measurements with a diurnal pattern.

40. Table S2: Why does the "Measurement number" jump?

**Response:** Thank you for your correction. After reanalysis all of the raw data and all postprocessing data (measurements 1-10), the number of peak samples did not change before and after postprocessing. Therefore, this table is no longer used in our revised manuscript. The detail information about it, please refer to line 356-357 for further confirmation.

41. Figure S3: In order to improve the visibility of the curve, you can consider reducing the thickness of the curve.

**Response**: Thank you for your suggestion. We have revised Fig. S3 (Fig. S6 after revised) to improve the visibility of the curve. Please refer to supporting information for further confirmation.

---

## Referee Report (RR1)

General:

- The English can still be improved, my previous comments on the readability, unfortunately, still hold.
- Please be strict with your use of the word "phenomena/phenomenon"

Abstract:

- Check the grammar

Introduction:

- Again, check grammar, sentence structures and lengths, and jargon.
- Specify that this study is investigates the effectiveness of noise-reduction algorithms available in the Aethlabs Dashboard. This way, users who do not use this dashboard understand why these algorithms were the ones investigated.
- Line 55: "Hagler" not "Hegler"
- The new paragraph (Lines 54-66) is better suited after the paragraph ending in line 105.
- I suggest this flow for the introduction:

BC definition and importance → BC high spatial variability (disadvantage of fixed stations) → portable instruments = mobile monitoring → introduce MA200 → challenges of MA200 and mobile monitoring (noise) → introduce existing noise reduction algorithm (ONA) and disadvantages → introduce Aethlabs dashboard and offered noise reduction algorithms → objective of the study

- I am not completely satisfied with the motivation. You mentioned the evaluation of ONA by Hagler (who actually created it) and Van den Bossche 2015 (who, if I'm not mistaken, did not evaluate ONA, merely applied it to their dataset). Between these references you have in the introduction, there is not enough evidence that the ONA did not perform well for mobile measurement datasets. I suggest the paper of Cheng et al., 2013 (10.4209/aaqr.2012.12.0371) although not for mobile applications. But to simplify things, you may motivate your study by stating that a full assessment on noise reduction algorithms for the new microaethalometers used for mobile monitoring haven't been fully assessed yet, etc. You may cite several mobile measurement studies who applied ONA on their datasets, but did not fully evaluate the effects of such data treatment. The telling of this story can still be improved.

Methodology:

- I think you don't need to mention the manufacturer again here. You already did in the introduction.
- Strictly speaking, the microaethalometers are absorption (or better, attenuation) photometers. The MA200 particularly only measures "equivalent black carbon" at 1 wavelength (880nm).
- Lines 125-132 belong in the introduction. In this part, you can just focus on the instruments' technical information relevant to your study.
- Line 135 has no relevance to the sentences that follow which are focused on quality assurance of the MA200 in the field. This subsection (2.1) can be rearranged:
  - Technical info of instrument and then quality assurance
  - Leave out the noise reduction algorithms part and move it to study design subsection because the analysis of these algorithms is the heart of this work.

- o Introduce here that you will only be using absorption/attenuation at one wavelength of the MA200.
- ➢ Section 2.3 can be combined with the quality assurance part of Section 2.1.
- ➢ Section 2.4: emphasize that these algorithms are the ones offered in the Aethlabs dashboard
- ➢ Line 206, specify with instrument Hagler used.
- ➢ Line 201, I think you mean Fig. S2 here.

Results and discussion

- ➢ I still do not understand the difference of NV (proportion of negative values) and NR (average noise reduction). In the methods, NR is defined as the # of negative values (after noise reduction) / total sample size. How is it different then from NV after noise reduction (which I believe is calculated the same way). And what is "average noise reduction", which part of it is averaged? Averaged over all measurements (1-10)? But then, reading line 349, the 72% and 87.4% is now the average reduction of peak samples? This is very confusing. Based on the caption of Table 2, it seems to me that it is only about the negative values, and not the peak samples. Either separate these two criteria or improve the table caption.
- ➢ Lines 383-396 are better suited in the methods part when talking about quality assurance.
- ➢ Figure S8, I think (b) is 10-s and (c) is 30s.

---

## Author Response (AR2)

Review of the revised version of amt-2020-517

Thank you very much for your consideration of our manuscript (amt-2020-517). We consider the comments from you very constructive and would like to thank you for the fine effort. Accordingly, we have made careful modifications. After taking a look into the house standards of the journal, we have revised the manuscript according to the guidelines. The revised manuscript has been reorganized, proofread by a language professional (Oxford English, as recommended), and marked in blue. The following are our detailed responses to each comment. We hope that the current version of the manuscript is qualified for publication in ***Atmospheric Measurement Techniques.***

**#Reviewer 1**

General:

1. The English can still be improved, my previous comments on the readability, unfortunately, still hold.

**Response**: We have carefully gone through the whole manuscript again to avoid grammatical errors. The full text of the manuscript has been checked and modified by a professional native English speaker to improve the readability of the manuscript and avoid misleading sentences. The terminology which was unsuitable for the study has been modified following previously published papers. We hope that the current version of the manuscript is qualified for publication.

2. Please be strict with your use of the word "phenomena/phenomenon"

**Response**: Thanks for your reminder. We have carefully modified it. Please refer to our revised manuscript for further confirmation.

Abstract:

1. Check the grammar

**Response**: We have carefully gone through the whole manuscript again to avoid grammatical errors. The full text of the manuscript has been checked and modified by a professional native English speaker to improve the readability of the manuscript and avoid misleading sentences. The terminology which was unsuitable for the study has been modified following previously published papers. We hope that the current version of the manuscript is qualified for publication.

Introduction:

1. Again, check grammar, sentence structures and lengths, and jargon.

**Response**: We have carefully gone through the whole manuscript again to avoid grammatical errors. The full text of the manuscript has been checked and modified by a professional native English speaker to improve the readability of the manuscript and avoid misleading sentences. The terminology which was unsuitable for the study has been modified following previously published papers. We hope that the current version of the manuscript is qualified for publication.

2. Specify that this study is investigates the effectiveness of noise-reduction algorithms available in the Aethlabs Dashboard. This way, users who do not use this dashboard understand why these algorithms were the ones investigated.

**Response**: Thank you for your correction. Nowadays, the portable microAeth® MA200 (MA200) is widely applied for measuring black carbon in human exposure characterization and mobile air quality monitoring. However, the field lacks information about this instrument's performance under various

settings. Therefore, it is important to provide this research – an evaluation of the real-time performance of the MA200 in an urban area – for MA200 users. Following this, our research also provides information about post-processing methods to non-users of MA200, which may also have applicability for other instruments.

3. Line 55: "Hagler" not "Hegler"
**Response**: Thanks for your correction. We have modified it. Please refer to line 81 for further confirmation.

4. The new paragraph (Lines 54-66) is better suited after the paragraph ending in line 105.
**Response**: Thanks for your correction. We have modified it according to your comment. Please refer to line 79-92 for further confirmation.

5. I suggest this flow for the introduction:
BC definition and importance → BC high spatial variability (disadvantage of fixed stations) → portable instruments = mobile monitoring → introduce MA200 → challenges of MA200 and mobile monitoring (noise) → introduce existing noise reduction algorithm (ONA) and disadvantages → introduce Aethlabs dashboard and offered noise reduction algorithms → objective of the study
**Response**: Thanks for your suggestion. We have reorganized the introduction section and modified it following your comment. Please refer to the introduction section in our revised manuscript for further confirmation.

6. I am not completely satisfied with the motivation. You mentioned the evaluation of ONA by Hagler (who actually created it) and Van den Bossche 2015 (who, if I'm not mistaken, did not evaluate ONA, merely applied it to their dataset). Between these references you have in the introduction, there is not enough evidence that the ONA did not perform well for mobile measurement datasets. I suggest the paper of Cheng et al., 2013 (10.4209/aaqr.2012.12.0371) although not for mobile applications. But to simplify things, you may motivate your study by stating that a full assessment on noise reduction algorithms for the new microaethalometers used for mobile monitoring haven't been fully assessed yet, etc. You may cite several mobile measurement studies who applied ONA on their datasets, but did not fully evaluate the effects of such data treatment. The telling of this story can still be improved.
**Response**: Thanks for your suggestion. We have modified it following your comment. Please refer to line 79-92 for further confirmation.
Methodology:
1. I think you don't need to mention the manufacturer again here. You already did in the introduction.
**Response**: Thanks for your suggestion. We have deleted the information of the manufacturer according to your comment.

2. Strictly speaking, the microaethalometers are absorption (or better, attenuation) photometers. The MA200 particularly only measures "equivalent black carbon" at 1 wavelength (880nm).

**Response**: Thanks for your suggestion. We have modified and integrated it according to your comment. Please refer to lines 102-103 for further confirmation.

3. Lines 125-132 belong in the introduction. In this part, you can just focus on the instruments' technical information relevant to your study.

**Response**: Thanks for your suggestion. We have deleted the part that is not relevant to the technical information of the instruments in our study. Please refer to subsection 2.1.1 in our revised manuscript for further confirmation.

4. Line 135 has no relevance to the sentences that follow which are focused on quality assurance of the MA200 in the field. This subsection (2.1) can be rearranged:
A. Technical info of instrument and then quality assurance.
B. Leave out the noise reduction algorithms part and move it to study design subsection because the analysis of these algorithms is the heart of this work.
C. Introduce here that you will only be using absorption/attenuation at one wavelength of the MA200.

**Response**: Thanks for your constructive suggestion. We have restructured and modified subsection 2.1 according to your comment. Please refer to subsection 2.1 in our revised manuscript for further confirmation.

5. Section 2.3 can be combined with the quality assurance part of Section 2.1.

**Response**: Thanks for your suggestion. We have combined and modified it according to your comment. Please refer to subsection 2.1 in our revised manuscript for further confirmation.

6. Section 2.4: emphasize that these algorithms are the ones offered in the Aethlabs dashboard

**Response**: Thanks for your suggestion. We have emphasized and added all algorithms for noise reduction offered in the Aethlabs dashboard. Please refer to section 2.3 (section 2.4 before) in our revised manuscript for further confirmation.

7. Line 206, specify with instrument Hagler used.

**Response**: Thanks for your suggestion. We have added the relating instruments that Hagler used. Please refer to line 192 for further confirmation.

8. Line 210, I think you mean Fig. S2 here.

**Response**: Thanks for your correction. We have checked and confirmed that this line described a lower $\Delta$ATN threshold of 0.01 for the mobile measurement data, which referred to Figure S3.

Results and discussion

1. I still do not understand the difference of NV (proportion of negative values) and NR (average noise reduction). In the methods, NR is defined as the # of negative values (after noise reduction) / total sample size. How is it different then from NV after noise reduction (which I believe is calculated the same way). And what is "average noise reduction", which part of it is averaged? Averaged over all measurements (1-10)? But then, reading line 349, the 72% and 87.4% is now the average reduction of peak samples? This is very confusing. Based on the caption of Table 2, it seems to me that it is only about the negative values, and not the peak samples. Either separate these two criteria or improve the table caption.

**Response**: Sorry for the confusion caused, the proportion of negative values (NV) remaining was calculated as the number of negative values divided by the total sample size, however, **the average noise reduction (NR) refers to the reduction of peak samples** and is calculated by the number of peak samples before post-processing ($C_i$) minus the number of peak samples after post-processing ($C_j$), and the difference value ($\triangle C = C_i - C_j$) was obtained. Then the change in the number of peak samples was divided by the total number of peak samples before post-processing data. In order to avoid misunderstanding by the readers, we changed **the average noise reduction (NR)** to **the average reduction of peak samples (RP)** in the Table 2 caption. Following this, we have revised this term in the whole manuscript. Please refer to our new Table 2 caption in our revised manuscript for further confirmation.

2. Lines 383-396 are better suited in the methods part when talking about quality assurance.

**Response**: Thanks for your suggestion. We have modified it according to your comment. Please refer to subsection 2.1, line 107-115, in our revised manuscript for further confirmation.

3. Figure S8, I think (b) is 10-s and (c) is 30s.

**Response**: Thanks for your correction. We have modified Figure S8 caption following your comment. Please refer to Fig.S8 caption in the supporting information for further confirmation.